# INT vs. FP: A Comprehensive Study of Fine-Grained Low-bit Quantization Formats

**Mengzhao Chen** [1 2]  **Meng Wu** [3]  **Hui Jin** [2]  **Zhihang Yuan** [2]  **Jing Liu** [2]  **Chaoyi Zhang** [2]  **Yunshui Li** [2]  **Jie Huang** [2]
**Jin Ma** [2]  **Zeyue Xue** [1]  **Zhiheng Liu** [1]  **Xingyan Bin** [2]  **Ping Luo** [1]

## Abstract

Modern AI hardware, such as Nvidia's Blackwell architecture, is increasingly embracing low-precision floating-point (FP) formats to handle the pervasive activation outliers in Large Language Models (LLMs). Despite this industry trend, a unified comparison of FP and integer (INT) quantization across varying granularities has been missing, leaving algorithm and hardware co-design without clear guidance. This paper fills that gap by systematically investigating the trade-offs between FP and INT formats. We reveal a critical performance crossover: while FP excels in coarse-grained quantization, the comparison at fine-grained (block-wise) levels is more nuanced. Our comprehensive comparison demonstrates that for popular 8-bit fine-grained formats (e.g., MX with block size 32), MXINT8 is superior to its FP counterpart in both algorithmic accuracy and hardware efficiency. However, for 4-bit formats, FP (e.g., MXFP4, NVFP4) often holds an accuracy advantagea, though we show that NVINT4 can surpass NVFP4 when outlier-mitigation techniques like Hadamard rotation are applied. We also introduce a symmetric clipping method that resolves gradient bias in fine-grained low-bit INT training, enabling nearly lossless performance for MXINT8 training. These findings challenge the current hardware trajectory, demonstrating that a one-size-fits-all FP approach is suboptimal and advocating that fine-grained INT formats, particularly MXINT8, offer a better balance of accuracy, power, and efficiency for future AI accelerators. Code is available at https://github.com/ChenMnZ/INT_vs_FP.

---

[1]The University of Hong Kong [2]ByteDance Seed [3]PicoHeart. Correspondence to: Xingyan Bin <binxingyan@bytedance.com>, Ping Luo <pluo@hku.hk>.

*Proceedings of the 43rd International Conference on Machine Learning*, Seoul, South Korea. PMLR 306, 2026. Copyright 2026 by the author(s).

## 1. Introduction

The proliferation of Large Language Models (LLMs) has been accompanied by a surge in their computational and memory demands (Yuan et al., 2024), making quantization an indispensable technique for efficient deployment. A central challenge in quantizing LLMs, particularly those based on the Transformer architecture, is the presence of significant outliers (Sun et al., 2024; Dettmers et al., 2022) in activation distributions. These outliers, characterized by their large magnitude but infrequent occurrence, pose a considerable problem for low-precision representations. To accommodate this wide dynamic range, the AI hardware industry (NVIDIA Corporation, 2024a) is increasingly pivoting towards low-precision floating-point (FP) formats, such as FP8 and FP4. Prominent examples like NVIDIA's Blackwell architecture (NVIDIA Corporation, 2024a) underscore this trend, favoring the superior dynamic range of FP to handle outliers more gracefully than traditional integer (INT) formats.

However, this industry-wide momentum towards FP formats is based on an incomplete picture. The comparative advantages of FP and INT have not been systematically evaluated across different quantization granularities in a unified framework. Most studies (Xiao et al., 2023; Chen et al., 2024a; Liu et al., 2024b) focus on a single format or compare them only at coarse granularities (e.g., per-channel), failing to answer a critical question: how does the performance trade-off between INT and FP evolve as granularity becomes finer? Since fine-grained (block-wise) quantization is now a standard technique (Rouhani et al., 2023; NVIDIA Corporation, 2024b) for mitigating outliers, understanding its interaction with the underlying number format is essential for effective algorithm-hardware co-design.

In this paper, we conduct a comprehensive, systematic comparison of fine-grained INT and FP quantization. Our investigation reveals a critical "crossover point" in performance. While FP formats hold a distinct advantage in coarse-grained scenarios, we find that INT formats become highly competitive as the block size shrinks, though the benefit depends heavily on the bit width. As granularity becomes finer, the local dynamic range within each block

is reduced, allowing the uniform precision of INT formats to become more effective. This trend is analyzed across modern block-wise formats, such as the 32-element blocks in Microscaling (MX) formats or the 16-element blocks in NVIDIA's (NV) formats. To enable a direct comparison, we introduce and evaluate integer variants (e.g., MXINT8, MXINT6, MXINT4, NVINT4) alongside their standard FP counterparts (e.g., MXFP8, MXFP6, MXFP4, NVFP4).

Our key contributions are as follows:

- We develop a theoretical and statistical framework that models the quantization signal-to-noise ratio (QSNR) for both INT and FP formats. This framework enables a direct theoretical comparison of their performance trade-offs and clarifies the crossover points and .

- We demonstrate that MXINT8 consistently outperforms MXFP8 in both direct-cast inference and low-bit training. We also show that NVINT4 can surpass NVFP4 when combined with Hadamard rotation. Critically, we introduce a symmetric clipping method that resolves a gradient bias, enabling nearly lossless MXINT8 low-bit training.

- We present a comparative hardware cost analysis, demonstrating that fine-grained INT formats are significantly more area and energy-efficient than their floating-point counterparts at matched throughput.

- Collectively, our findings challenge the prevailing FP-centric trajectory in AI hardware design and advocate for prioritizing fine-grained INT formats to achieve a more optimal balance of accuracy and efficiency in future AI accelerators.

## 2. Related Work

**Quantization Algorithms.** Quantization methods include post-training quantization (PTQ) (Lin et al., 2023; Frantar et al., 2022; Shao et al., 2023; Xiao et al., 2023) and quantization-aware training (QAT) (Chen et al., 2024b; Liu et al., 2025), which speed up inference. Low-bit training (Mishra et al., 2025; Tseng et al., 2025; Chen et al., 2025b) speeds up both training and inference. Several works also study scaling laws (Hoffmann et al., 2022) for low-bit quantization (Castro et al., 2025; Chen et al., 2025a; Frantar et al., 2025; Kumar et al., 2024). However, most prior work focuses on a single low-bit format—either integer or floating-point—and does not provide direct comparisons between these formats. Recently, MR-GPTQ (Egiazarian et al., 2025) finds that Hadamard rotation (Castro et al., 2025) improves the accuracy of MXFP4 but hurts the accuracy of NVFP4. In this paper, we support this finding with both experiments and theory.

**INT vs. FP.** Several works compare integer and floating-point formats. (Lee et al., 2023; Zhang et al., 2024b) study mixed-format quantization in the PTQ setting and assign integer or floating-point formats to different model layers. SageAttention (Zhang et al., 2024a) observes that INT8 quantization for $QK^\top$ in self-attention achieves higher accuracy than FP8 formats, and SageAttention3 (Zhang et al., 2025) shows that, in the backward pass of 8-bit attention, INT8 outperforms FP8 in accuracy. (Kuzmin et al., 2022; Van Baalen et al., 2023) finds that FP8 performs better under outlier-heavy distributions, while INT8 performs better under Gaussian-like distributions. Most existing work compares integer and floating-point formats empirically. In this paper, we provide a clearer quantitative and theoretical comparison, and our analysis generalizes to different bit-widths, quantization block sizes, exponent bits in floating-point formats, and data distributions.

**Hardware.** Previous accelerators (NVIDIA Corporation, 2020; 2022) do not natively support fine-grained quantization, so algorithms (Xiao et al., 2023; Chen et al., 2024a) face challenges with per-channel quantization in the presence of outliers (Sun et al., 2024). Recently, OCP (Rouhani et al., 2023) proposes Microscaling (MX) data formats, which combine a per-block scaling factor with a block size of 32 to improve low-bit quantization performance. NVIDIA Blackwell (NVIDIA Corporation, 2024a) supports MXFP8, MXFP4, and NVFP4 at the hardware level.

## 3. Preliminaries

Quantization maps a high-precision tensor $\mathbf{X}$ to a lower bit-width. In this section, we present low-bit integer (INT) quantization, floating-point (FP) quantization, quantization granularity with a focus on fine-grained block-wise schemes, and an overview of existing low-bit block formats.

### 3.1. Low-Precision Integer Formats

For $b$-bit integer quantization, we define:

$$\mathbf{X_q} = \text{clip}\left(\left\lfloor \frac{\mathbf{X}}{s} \right\rceil, Q_{\min}, Q_{\max}\right) \cdot s, \qquad (1)$$

where $s$ is the scale factor that normalizes $\mathbf{X}$ to the target integer range, $\lfloor \cdot \rceil$ is round-to-nearest, and $\mathbf{X_q}$ is the dequantized tensor. The clipping ensures that the integer values lie in $[Q_{\min}, Q_{\max}]$ (e.g., for signed $b$-bit integers, $Q_{\min} = -2^{b-1}$ and $Q_{\max} = 2^{b-1} - 1$).

### 3.2. Low-Precision Floating-Point Formats

Floating-point representation (Markstein, 2008) uses three fields: the sign bit ($S$), the exponent ($E$), and the mantissa ($M$). We denote a format as E$x$M$y$, where $x$ and $y$ are the numbers of exponent and mantissa bits. The sign de-

termines the polarity, the exponent sets the dynamic range, and the mantissa sets the precision. A floating-point number decodes as:

$$\mathbb{C}_{\text{FP}} = \begin{cases} (-1)^s \times (1.m)_2 \times 2^{e-\text{bias}}, \\ \quad \text{if } e \neq 0 \text{ (Normal)}, \\ (-1)^s \times (0.m)_2 \times 2^{1-\text{bias}}, \\ \quad \text{if } e = 0, \, m \neq 0 \text{ (Subnormal)}. \end{cases} \quad (2)$$

where $s$, $e$, and $m$ are the sign, exponent and mantissa values of a floating-point number. Hence, $\mathbb{C}_{\text{FP}}$ denotes the set of representable low-bit floating-point values. Floating-point quantization is:

$$\mathbf{X_q} = \text{Nearest}\left(\frac{\mathbf{X}}{s}, \mathbb{C}_{\text{FP}}\right) \cdot s, \quad (3)$$

where $\text{Nearest}(\cdot, \mathbb{C}_{\text{FP}})$ maps normalized values to the nearest element of $\mathbb{C}_{\text{FP}}$. Eq. (3) is a general quantization form that also recovers integer quantization by replacing $\mathbb{C}_{\text{FP}}$ with $\mathbb{C}_{\text{INT}}$.

### 3.3. Quantization Granularity

Quantization granularity specifies how scale factors apply across a tensor. Finer granularity usually improves accuracy but increases compute and memory overhead due to more scale factors. Common choices are: (i) Per-tensor: a single scale for the entire tensor. (ii) Per-channel: a scale per channel, broadcast along a chosen axis. (iii) Block-$k$: the tensor is partitioned into $1 \times k$ blocks along one dimension, and each block has its own scale. Block quantization is a key technique for improving accuracy at low precision. In this paper, we mainly focus on block quantization.

### 3.4. Block-Quantization Formats

To improve low-bit accuracy, OCP (Rouhani et al., 2023) proposes the Microscaling (MX) format, which uses a shared UE8M0[1] scale for each block of 32 elements. This fine-grained scaling reduces quantization error. Recently, NVIDIA Blackwell-series GPUs (NVIDIA Corporation, 2024b) provide native hardware support for MXFP8/MXFP6/MXFP4. Traditionally, FP8 has E4M3 and E5M2 variants, and FP6 has E2M3 and E3M2 variants. We consider E4M3 for MXFP8 and E2M3 for MXFP6 because mantissa bits are more critical to the performance of fine-grained quantization, consistent with prior work (Liu et al., 2024a; Mishra et al., 2025; Rouhani et al., 2023). Furthermore, NVIDIA proposes NVFP4, which enhances MXFP4 by reducing the block size from 32 to 16 and replacing the UE8M0 scale with an E4M3 scale. NVFP4 also introduces

---

[1] UE8M0 is an 8-bit unsigned floating-point format with eight exponent bits and zero mantissa bits.

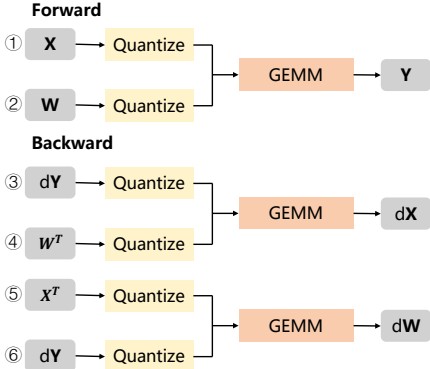

*Figure 1.* Compute flow of low-bit forward and backward propagation of linear layer.

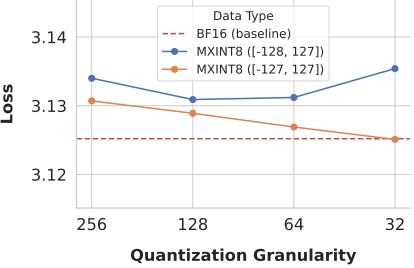

*Figure 2.* Impact of clipping range on INT8 final training loss on 145M model with 20B training tokens. Scale factor is kept on BF16 to emphasize the harm of asymmetric representation space during low-bit training.

a second-level per-tensor scale to prevent overflow of the first-level E4M3 scale. Therefore, current hardware tends to support low-bit fine-grained floating-point formats. To enable fair comparison between low-bit floating-point and integer formats, we also introduce four corresponding integer variants: MXINT8, MXINT6, MXINT4, and NVINT4. Details of these low-bit formats are listed in Table 1.

## 4. Quantization Recipe

This section illustrates the computation flow for low-bit inference and training in Sec. 4.1, and details the scale-factor computation used in quantization in Sec. 4.2.

### 4.1. Quantization Compute Flow

Figure 1 shows an example of using low-bit GEMM in a linear layer during forward and backward propagation. Given high-precision (e.g., BFloat16) activations $\mathbf{X}$ and weights $\mathbf{W}$, the forward pass of the quantized linear layer[2]

---

[2] We omit the bias term.

*Table 1.* Low-bit formats name and their correspond represented range and scale factors.

| Format | Block Size | Max Value | Min Value | Dynamic Range | Scale-1 | Scale-2 |
|---|---|---|---|---|---|---|
| MXFP8 (E4M3) | 32 | $\pm 448$ | $\pm 2^{-9}$ | $1.75 \times 2^{17}$ | UE8M0 | - |
| MXINT8 | 32 | 127 | 1 | 127 | UE8M0 | - |
| MXFP6 (E2M3) | 32 | $\pm 7.5$ | $\pm 0.125$ | 60 | UE8M0 | - |
| MXINT6 | 32 | $\pm 31$ | $\pm 1$ | 31 | UE8M0 | - |
| MXFP4 (E2M1) | 32 | $\pm 6$ | $\pm 0.5$ | 12 | UE8M0 | - |
| MXINT4 | 32 | $\pm 7$ | $\pm 1$ | 7 | UE8M0 | - |
| NVFP4 | 16 | $\pm 6$ | $\pm 0.5$ | 12 | E4M3 | FP32 |
| NVINT4 | 16 | $\pm 7$ | $\pm 1$ | 7 | E4M3 | FP32 |

is:

$$\mathbf{Y} = \underbrace{\text{Quantize}(\mathbf{X})}_{①} \underbrace{\text{Quantize}(\mathbf{W})}_{②}. \tag{4}$$

The backward pass to compute $d\mathbf{X}$ and $d\mathbf{W}$ is:

$$d\mathbf{X} = \underbrace{\text{Quantize}(\mathbf{dY})}_{③} \underbrace{\text{Quantize}(\mathbf{W}^T)}_{④}, \tag{5}$$

$$d\mathbf{W} = \underbrace{\text{Quantize}(\mathbf{X}^T)}_{⑤} \underbrace{\text{Quantize}(\mathbf{dY}^T)}_{⑥}. \tag{6}$$

Quantize(·) maps high-precision tensors to low-bit representations. Thus, there are six quantization operations in one linear layer: ① $\mathbf{X}$ and ② $\mathbf{W}$ in Eq. (4); ③ $\mathbf{dY}$ and ④ $\mathbf{W}^T$ in Eq. (5); ⑤ $\mathbf{X}^T$ and ⑥ $\mathbf{dY}^T$ in Eq. (6). Block-wise quantization requires tensors to be quantized along the GEMM reduction dimension to gain hardware benefits. Therefore, ① and ⑤, ② and ④, and ③ and ⑥ are quantized along different axes (Liu et al., 2024a; Darvish Rouhani et al., 2023). We separately analyze the quantization error of these six operations in Sec. 6.1.

### 4.2. Quantization Operation

**UE8M0 scale factor.** The scale factor $s$ in Eq. (1) and Eq. (3) is computed with the AbsMax quantizer:

$$s = \frac{\text{AbsMax}(\mathbf{X})}{Q_{max}}, \tag{7}$$

where AbsMax($\mathbf{X}$) is the maximum absolute value within the group of values that share a single scale factor, and $Q_{max}$ is the maximum value of the quantized type (see Table 1). Eq. (7) maps the largest magnitude in high precision to the maximum representable low-precision value without clipping. OCP (Rouhani et al., 2023) further converts the high-precision scale factor to the UE8M0 format for MX formats:

$$s' = 2^{\text{clip}(\lfloor \log_2(\text{AbsMax}(\mathbf{X})) \rfloor - \lfloor \log_2(Q_{max}) \rfloor, -127, 127)}, \tag{8}$$

where $\lfloor \cdot \rfloor$ denotes rounding down. Eq. (8) rounds the high-precision scale down to the nearest UE8M0 value,

which introduces extra clipping error. Following existing works (Tseng et al., 2025; Chen et al., 2025b; Mishra et al., 2025), we round up the UE8M0 scale based on Eq. (7) to avoid this error:

$$s' = 2^{\text{clip}(\lceil \log_2(s) \rceil, -127, 127)}, \tag{9}$$

where $\lceil \cdot \rceil$ denotes rounding up.

**Symmetric Clipping.** Floating-point formats are naturally symmetric around zero. In contrast, signed integers in two's complement have one extra negative value: for a $b$-bit integer, $Q_{min} = -2^{b-1}$ and $Q_{max} = 2^{b-1} - 1$ (NVIDIA Corporation, 2024b). We find that this asymmetric range usually does not affect inference. However, as shown in Figure 2, it degrades INT8 training due to a persistent negative bias in gradients. Finer-grained quantization suffers more because more values fall into the unique negative endpoint $Q_{min}$. For INT8, the minimum value in a group can still map to $-128$ even when the scale is set to AbsMax($\mathbf{X}$)/127 due to BFloat16 arithmetic precision (see Sec. C.2 for details). Therefore, we use a symmetric integer range for all INT quantizers as shown in Table 1:

$$Q_{min} = -(2^{b-1} - 1), \quad Q_{max} = 2^{b-1} - 1,$$

In this section, we analyze low-bit integer and floating-point formats and build a theoretical framework for comparing them. Section 5.1 derives theorems for the quantization signal-to-noise ratio (QSNR), and Section 5.2 compares low-bit formats based on the theoretical QSNR.

## 5. Theoretical Framework

### 5.1. Theoretical QSNR

**QSNR Metric.** We use the Quantization Signal-to-Noise Ratio (QSNR, dB) (Darvish Rouhani et al., 2023) to measure numerical fidelity under different quantization schemes. QSNR is the ratio of the power of the original signal $\mathbf{X}$ to the power of the quantization noise $\mathbf{X} - \mathbf{X}_q$, expressed in decibels:

$$\text{QSNR} = -10 \log_{10} \left( \frac{\|\mathbf{X} - \mathbf{X}_q\|^2}{\|\mathbf{X}\|^2} \right). \tag{10}$$

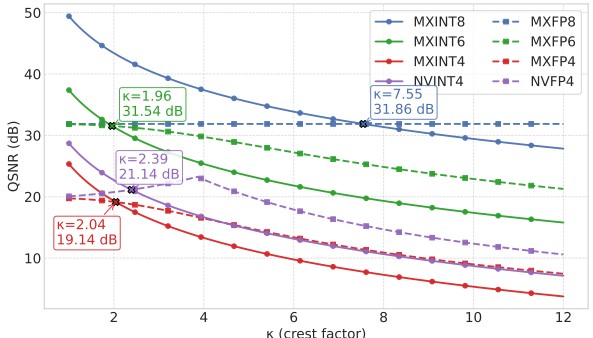

*Figure 3.* Theoretical QSNR comparison between various integer (INT) and floating-point (FP) formats across a range of crest factors ($\kappa$), derived from Eq. (13) and Eq. (14). The boxes represent the crest factor and QSNR of the crossover point of the INT and FP curves.

A higher QSNR means the quantized vector better preserves the magnitude and direction of the original vector.

**Common assumptions.** We consider block vectors $\mathbf{X} \in \mathbb{R}^k$ with i.i.d. entries $X_i \sim \mathcal{N}(0, \sigma^2)$. The block root-mean-square (RMS) equals $\sigma$, and the **crest factor** is

$$\kappa := \frac{\max(|\mathbf{X}|)}{\sigma}. \tag{11}$$

We use blockwise absolute-maximum (AbsMax) scaling:

$$s' = \rho \, s, \tag{12}$$

where $s$ is the high-precision scale from Eq. (7), and $\rho$ models the overhead of the low-precision scale. For example, the UE8M0 scale in Eq. (9) has $\rho \in [1, 2)$, while for the E4M3 scale in NV-format we set $\rho = 1$[3].

where $s$ is the high-precision scale from Eq. (7), and $\rho$ models the overhead of the low-precision scale. For example, the UE8M0 scale in Eq. (9) has $\rho \in [1, 2)$, while for the E4M3 scale in NV-format we set $\rho = 1$ since it is close to BFloat16 scales.

**Theorem 1 (INT QSNR).** Under $b$-bit INT quantization, the QSNR (in dB) is

$$\text{QSNR}_{\text{INT}} \approx \begin{cases} 4.78 + 6.02 \, b - 20 \log_{10}(\rho) - 20 \log_{10}(\kappa), \\ \quad \text{(UE8M0 scale)} \\ 4.78 + 6.02 \, b - 20 \log_{10}(\kappa) + 10 \log_{10}\left(\frac{g}{g-1}\right), \\ \quad \text{(E4M3 scale)} \end{cases} \tag{13}$$

---

[3]With 3 mantissa bits, E4M3 has a maximum relative gap of 1.125. Rounding up the scale to prevent clipping strictly bounds the overhead to $\rho \in [1, 1.125]$, justifying the $\rho = 1$ approximation.

A detailed proof of Theorem 1 appears in Sec. A.2, where $b$ is the bit width, $\rho$ is the scale overhead, $\kappa$ is the crest factor in Eq. (11), and $g$ is the block size.

**Interpretation of Theorem 1.** (i) Each extra bit gives $\approx 6.02$ dB. (ii) UE8M0 scaling incurs up to $20 \log_{10}(\rho) \leq 6.02$ dB loss. (iii) A larger crest factor $\kappa$ reduces QSNR; smaller blocks usually reduce $\kappa$ and improve QSNR. (iv) E4M3 scaling has no $\rho$ overhead and avoids the per-block maximum error, giving a $10 \log_{10}\left(\frac{g}{g-1}\right)$ QSNR gain.

**Theorem 2 (FP QSNR).** Under FP quantization, the QSNR (in dB) is

$$\text{QSNR}_{\text{FP}} \approx \begin{cases} -10 \log_{10}\left(\alpha_M \, w_{\text{norm}} \, + \, \beta \, (\rho \, \kappa)^2 \, p_{\text{sub}}\right), \\ \quad \text{(UE8M0 scale)} \\ -10 \log_{10}\left(\alpha_M \left(w_{\text{norm}} - \frac{\kappa^2}{g}\right) \, + \, \beta \, \kappa^2 \, p_{\text{sub}}\right), \\ \quad \text{(E4M3 scale)} \end{cases} \tag{14}$$

A detailed proof of Theorem 2 appears in Sec. A.3, with $\alpha_M = \frac{1}{24 \cdot 2^{2M}}$ (mantissa resolution term) and $\beta = \frac{2^{2(1-B-M)}}{12 \, Q_{\text{max}}^2}$. Here $M$ is the mantissa bit width, $B$ is the exponent bias, and $Q_{\text{max}}$ is the largest finite normal magnitude of the target FP format (e.g., $Q_{\text{max}} = 448$ for E4M3). The terms $w_{\text{norm}}$ and $p_{\text{sub}}$ measure how much of the distribution falls into the normal and subnormal regions (after scaling): $w_{\text{norm}}$ is the fraction of signal energy carried by normal FP numbers and incurs mantissa quantization error $\alpha_M$; $p_{\text{sub}}$ is the probability that a value encodes as subnormal and incurs a fixed absolute step error.

**Interpretation of Theorem 2.** (i) The mantissa bit width sets the upper bound on FP QSNR. With ample dynamic range ($w_{\text{norm}} \approx 1$ and $p_{\text{sub}} \approx 0$), QSNR $\approx 13.80 + 6.02 \, M$ dB, independent of block granularity and the distribution of $\mathbf{X}$. (ii) A larger crest factor $\kappa$ increases the share of subnormals and reduces QSNR. Finer-grained blocks reduce $\kappa$, lower $p_{\text{sub}}$, and improve QSNR. (iii) E4M3 scaling has no $\rho$ overhead and avoids the per-block maximum error, reducing $\frac{\kappa^2}{g}$ error energy in the normal region.

### 5.2. Theoretical Comparisons

With Eq. (13) in Theorem 1 and Eq. (14) in Theorem 2, we estimate the QSNR of low-bit integer and floating-point formats for a given bit width and target distribution (via $\kappa$). Specifically, we set $\rho = 1.5$ to imitate UE8M0 scale. As shown in Figure 3, we observe:

- **MXINT8 *vs.* MXFP8**: MXFP8 QSNR varies smoothly due to its ample dynamic range. MXINT8 outperforms FP8 when $\kappa < 7.55$.

- **MXINT6 *vs.* MXFP6**: MXFP6 has the same QSNR as

MXFP8 at small $\kappa$, because both MXFP6 and MXFP8 have three mantissa bits. However, FP6 QSNR decreases rapidly as $\kappa$ increases due to limited dynamic range. MXINT6 outperforms MXFP6 only when $\kappa < 1.96$.

- **MXINT4 *vs.* MXFP4**: MXINT4 outperforms MXFP4 when $\kappa < 2.04$.

- **NVINT4 *vs.* NVFP4**: NVINT4 outperforms NVFP4 when $\kappa < 2.39$. One interesting phenomenon is that NVFP4's QSNR even increase when $\kappa < 4$, this can be explained by Eq (14) that larger $\kappa$ can decrease the error of normal domain but increase the error of subnormal domain. In the relatively small $\kappa$ ($\kappa < 4$), normal domain dominate the error so that NVFP4' QSNR can increase when $\kappa < 4$.

Therefore, the key factor when comparing FP and INT formats is the data's crest factor $\kappa$.

# 6. INT *vs.* FP

We compare low-bit integer and floating-point formats at three levels. Section 6.1 analyzes the crest factor and QSNR for six types of intermediate tensors in Figure 1, offering a tensor-level perspective. Section 6.2 evaluates direct-cast inference, quantizing only the forward process. Section 6.3 presents results for low-bit training, quantizing both forward and backward processes.

## 6.1. Tensor-wise Analysis

**Setup.** To measure the QSNR in real data, we feed 8 Wiki-Text2 (Merity et al., 2016) sequences of length 4096 into Llama-3.1-8B, run both forward and backward propagation in BFloat16 precision, and capture the six intermediate tensors (weights, activations, and gradients) indicated by ①–⑥ in Figure 1. Llama-3.1-8B contains 224 linear layers across all transformer blocks. We collect these tensors for all 224 linear layers, leads totally $224 \times 6 = 10752$ tensors, and use them to compute the crest factors under different block size and QSNR under different low-bits formats. Specifically, QSNR is directly calculated tensor-wise, and crest factor is calculated block-wise and then average across the tensor. Additonally, we also apply random hadamard rotation (Ashkboos et al., 2024) ($32 \times 32$ for MX-format and $16 \times 16$ for NV-format) to measure the effectiveness of such outlier surpression technical to crest factor and QSNR.

**Crest factor results.** Table 2 reports crest factor statistics in boxplot form. We focus on the **75% quantile (*i.e.*, Q3)**, which reflects typical worst-case behavior across 75% of cases. For channel-wise quantization (block size $-1$), Q3 is 11.97, which is far above the crossover point in Figure 3. This indicates that FP outperforms INT in most cases

with coarse granularity. For the MX-format with block size 32, Q3 is 2.96. This value is well below the MXINT8 *vs.* MXFP8 crossover point (7.55), so MXINT8 outperforms MXFP8 in most cases. In contrast, 2.96 is above the MX-INT6 *vs.* MXFP6 and MXINT4 *vs.* MXFP4 crossover points (1.96 and 2.04), so MXINT6 and MXINT4 underperform their FP counterparts. After Hadamard rotation, Q3 decreases from 2.96 to 2.39, which remains below 7.55 but above 1.96 and 2.04; thus, MXINT8 still wins, while MX-INT6 and MXINT4 still lag behind MXFP6 and MXFP4. For the NV-format with block size 16, Q3 is 2.39, which equals the NVINT4 *vs.* NVFP4 crossover point and then decreases to 2.11 after Hadamard rotation, favoring NVINT4 over NVFP4 post-rotation.

**Crest factor vs. QSNR results.** Figure 4 reports measured QSNR across crest factors. The empirical trends closely follow the theoretical comparisons in Sec. 5 (Theorems 1–2) and the aforementioned crest factor reults:

- **MXINT8 *vs.* MXFP8**: The QSNR of MXFP8 is nearly constant at 31.50 because of its large dynamic range and mantissa-bit bound. MXINT8 has an average QSNR of 40.35, and thus significantly outperforms MXFP8.
- **MXINT6 *vs.* MXFP6 and MXINT4 *vs.* MXFP4**: MX-INT6 and MXINT4 consistently lag behind MXFP6 and MXFP4, with or without random Hadamard rotation.
- **NVINT4 *vs.* NVFP4**: Although the win rate of NVINT4 is $64.3\%$, its average QSNR is 20.55, which is slightly below NVFP4's 20.60 because NVINT4's QSNR decreases faster than NVFP4's as the crest factor increases. After random Hadamard rotation, NVINT4's average QSNR rises to 21.65, surpassing NVFP4's 20.35. Note that NVFP4's QSNR decreases from 20.60 to 20.35 after rotation, which is consistent with Figure 3: rotation reduces the crest factor, and when the crest factor is below 4, NVFP4's QSNR increases with the crest factor, so a reduction in crest factor lowers its QSNR.

Overall, real-data measurements corroborate the theory in Sec. 5.

## 6.2. Direct-Cast Inference

**Precisions.** For inference, we compare the formats in Table 1: MXFP8, MXINT8, MXFP6, MXINT6, MXFP4, MXINT4, NVFP4, and NVINT4. We perform direct-cast inference from a pretrained BFloat16 model and quantize all forward GEMMs.

**Models.** We evaluate 12 LLMs covering dense and Mixture-of-Experts (MoE) architectures, from 0.6B to 235B parameters: Qwen3-0.6B/1.7B/4B/8B/14B/32B/30B-A3B/235B-A22B (Yang et al., 2025), Llama-3.1-8B/70B, and Llama-3.2-1B/3B (Dubey et al., 2024). We also apply random

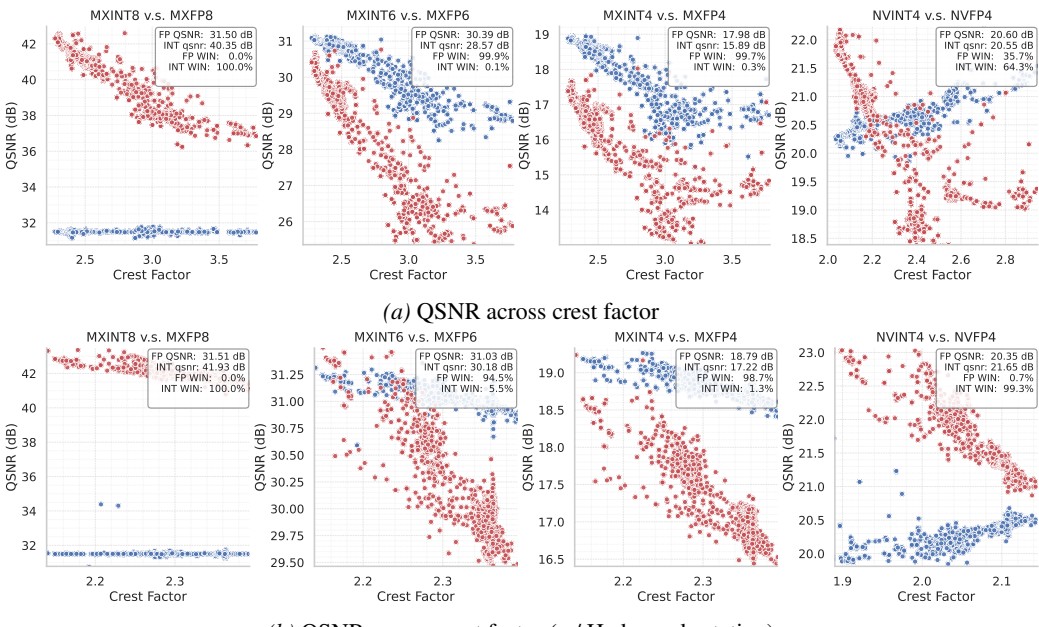

*(a)* QSNR across crest factor

*(b)* QSNR across crest factor (w/ Hadamard rotation)

*Figure 4.* Practical QSNR across crest factors from 10752 tensors source from ① to ⑥ in compute flow in Figure 1. (a) is the results from vanilla tensor and (b) applies random hadamard rotation to the tensor before quantization. The box in top right report the average QSNR of INT and FP quantization, and the win rates of INT and FP quantization.

*Table 2.* Summary statistics of the crest factor by block size in boxplot form. Q1 and Q3 denote the 25% and 75% quantiles, respectively.

| Type | Block Size | Min | Q1 | Median | Q3 | Max |
|------|-----------|-----|-----|--------|-----|-----|
| | -1 | 3.55 | 4.26 | 6.2 | 11.97 | 60.15 |
| Crest factor | 32 | 2.28 | 2.40 | 2.48 | 2.96 | 4.26 |
| | 16 | 2.04 | 2.13 | 2.16 | 2.39 | 3.16 |
| | -1 | 3.62 | 3.9 | 4.15 | 5.79 | 13.02 |
| Crest factor w/ hadamard rotation | 32 | 1.91 | 2.29 | 2.35 | 2.36 | 2.57 |
| | 16 | 1.77 | 2.06 | 2.1 | 2.11 | 2.21 |

*Table 3.* **Direct-cast inference comparisons** across 12 models. RHT denotes random Hadamard rotation. Per-model numbers appear in the Appendix.

| | Original | | w/ RHT | |
|---|---|---|---|---|
| | INT Win | FP Win | INT Win | FP Win |
| MXINT8 v.s. MXFP8 | **12** | 0 | **12** | 0 |
| MXINT6 v.s. MXFP6 | 0 | **12** | 1 | **11** |
| MXINT4 v.s. MXFP4 | 0 | **12** | 0 | **12** |
| NVINT4 v.s. NVFP4 | 0 | **12** | **12** | 0 |

Hadamard rotation and quantize $\mathbf{XR}$ and $\mathbf{R}^\top\mathbf{W}$, where $\mathbf{R}$ is a random Hadamard matrix of size $h \times h$. We set $h$ to the block size (32 for MX formats and 16 for NV formats). We provide official open-source links in Sec. C.

**Metrics.** Our goal is to compare integer and floating-point low-bit formats under the same settings, so ranking is more informative than absolute accuracy. Following (Dutta et al., 2024), accuracy alone is not sufficient for compressed mod-

els because it can hide large behavioral changes. We therefore use distance metrics: specifically, we compute the KL divergence on WikiText2 (Merity et al., 2016) between each quantized model and its BFloat16 counterpart. To reduce noise, we compute the divergence over the softmax distribution restricted to the top-25 logits of the BFloat16 model.

**Results.** Table 3 summarizes the comparison between FP and INT formats. Without rotation, MXINT8 outperforms MXFP8 on all 12 models, while MXINT6, MXINT4, and NVINT4 perform worse than MXFP6, MXFP4, and NVFP4. Although NVINT4 and NVFP4 have similar average QSNR in Figure 4a, NVINT4 loses more often because higher crest factors create more worst-case behavior for integers. With random Hadamard rotation, MXINT8 and NVINT4 win on all 12 models; MXINT6 wins 1 of 12 and MXINT4 loses all 12, consistent with the tensor-wise analysis in Sec. 6.1.

*Table 4.* **Low-bit training comparisons.** HS, OB, and WG represents Hellaswag, OpenbookQA, and WinoGrande, respectively.

| Model size | Training tokens | Precision | loss | Arc_E | Arc_C | HS | OB | PIQA | WG | Avg. |
|---|---|---|---|---|---|---|---|---|---|---|
| 1B | 100B | BF16 | 2.6727 | 37.80 | 69.40 | 60.20 | 38.40 | 74.43 | 61.09 | 56.89 |
| 1B | 100B | MXFP8 | 2.6767 | 37.03 | 69.82 | 60.28 | 38.00 | 74.37 | 61.64 | 56.86 |
| 1B | 100B | MXINT8 | **2.6758** | 37.95 | 69.45 | 60.02 | 38.80 | 74.54 | 61.38 | **57.02** |
| 3B | 200B | BF16 | 2.4794 | 46.50 | 75.42 | 72.28 | 45.00 | 78.07 | 69.45 | 64.45 |
| 3B | 200B | MXFP8 | 2.4821 | 46.70 | 74.12 | 72.08 | 44.60 | 77.56 | 69.25 | 64.05 |
| 3B | 200B | MXINT8 | **2.4812** | 46.10 | 75.58 | 72.00 | 44.80 | 77.78 | 69.55 | **64.30** |

*Table 5.* Normalized energy and area costs of low-bit formats at same throughput. Single-format results use MXFP8 as the baseline, and mixed-format results use MXFP8+NVFP4 as the baseline.

| | Single Format | | | | Mixed Format | |
|---|---|---|---|---|---|---|
| | MXFP8 | MXINT8 | NVFP4 | NVINT4 | MXFP8+NVFP4 | MXINT8+NVINT4 |
| Energy | 1x | **0.63x** | 0.55x | **0.34x** | 1x | **0.75x** |
| Area | 1x | **0.79x** | 0.54x | **0.38x** | 1x | **0.66x** |

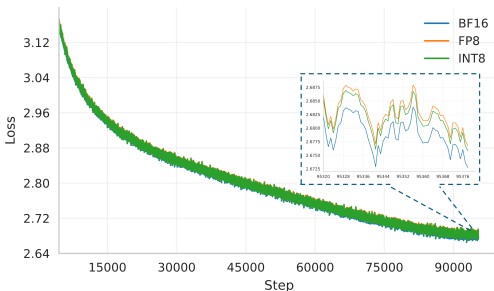

*Figure 5.* Loss curves comparison among BF16, MXFP8 and MXINT8 training on Llama-1B with 100B tokens. Results are smoothed by exponential moving average with a coefficient of 0.9.

### 6.3. Training

**Precisions.** For training, we focus on nearly lossless low-bit training, which is more practical. Therefore, we study only the 8-bit setting and compare MXINT8 and MXFP8, since FP8 training is demonstrated to be nearly lossless in prior work (Mishra et al., 2025; Liu et al., 2024a).

**Models and datasets.** We train 1B and 3B Llama3-style (Dubey et al., 2024) models on the OLMo2-Mix-1124 (OLMo et al., 2024) pretraining dataset, with 100B and 200B training tokens, respectively. Detailed model architectures and training hyperparameters are in Sec. C.

**Metrics.** We measure training performance using two metrics: training loss and task accuracy. We smooth the training loss with an exponential moving average (coefficient 0.9). We compute all accuracies with `lm_eval` (Gao et al., 2024) through 5-shot evaluation. We report `acc` for WinoGrande (Sakaguchi et al., 2021) and `acc_norm` for HellaSwag (Zellers et al., 2019), Arc_Challenge, Arc_Easy (Clark et al., 2018), PIQA (Bisk et al., 2020),

and Openbookqa (Mihaylov et al., 2018).

**Results.** Figure 5 shows the loss curves for BF16, MXFP8, and MXINT8 training. The curves for MXFP8 and MXINT8 almost overlap with BF16. In addition, MXINT8 consistently outperforms MXFP8 with a loss that is lower by approximately 0.001, as shown in the enlarged view in Figure 5. Table 4 shows that MXINT8 also achieves nearly the same average accuracy across six common-sense reasoning tasks compared to BF16 training. These results demonstrate that MXINT8 supports nearly lossless low-bit training, while existing works (Liu et al., 2024a; Mishra et al., 2025) mainly focus on FP8 training.

## 7. Hardware Cost Analysis

Based on the hardware model in Sec. B, we evaluate the energy and area cost of a Matrix-Multiply Unit (MMU) that supports the MX format. Table 5 shows that MXINT8 and NVINT4 reduce energy by 37% and 38%, respectively, compared with MXFP8 and NVFP4. We also evaluate mixed-format configurations. Following the NVIDIA Blackwell GPUs (NVIDIA Corporation, 2024b), we study a chip that supports both 8-bit and 4-bit data types and set the throughput ratio of 8-bit to 4-bit to 1:2 to match the communication bandwidth. As shown in Table 5, the "MXINT8+NVINT4" configuration further reduces area by about 34% relative to "MXFP8+NVFP4", mainly because circuit reuse is simpler in the INT pipeline (Table 7). Overall, this analysis shows that, at matched throughput, low-bit integer formats are more hardware-efficient than low-bit floating-point formats.

## 8. Conclusion

Our comprehensive study reveals a critical and nuanced trade-off between integer (INT) and floating-point (FP) quantization. We find that while FP formats are effective

at coarse granularities, the popular fine-grained MXINT8 consistently outperforms its FP counterpart MXFP8 in both accuracy and hardware efficiency. For 4-bit formats, the accuracy advantage shifts to FP (MXFP4, NVFP4) , though we demonstrate that NVINT4 can surpass NVFP4 when combined with random Hadamard rotation. These findings challenge the current hardware trajectory, which is increasingly focused on FP. We therefore call for a strategic shift in both academia and industry toward algorithm-hardware co-design that re-evaluates and prioritizes fine-grained INT formats to build more powerful and efficient AI accelerators.

## 9. Limitations

While this study systematically compares INT and FP formats, our analysis framework has several limitations that present opportunities for future work:

- **Scope of Fine-Grained Quantization:** Our conclusion that INT outperforms FP is specific to fine-grained blocks (e.g., size 16 or 32). In coarse-grained settings (e.g., per-channel), high crest factors dictate that FP remains strictly superior.

- **Omission of Global Scaling Error:** The current QSNR model mathematically bounds but does not explicitly formulate the quantization error introduced by second-level global scaling (e.g., the E4M3 global scale in NV formats), focusing primarily on block-level scales.

- **Theoretical Assumptions:** Our derivations assume i.i.d. Gaussian inputs and rely on Bennett's high-resolution quantization model. In practice, LLM activations can be heavy-tailed, and Bennett's core assumptions may weaken at extreme low bit-widths (e.g., 4-bit).

- **Incompatibility with Adaptive Scale Clipping:** Our current framework accurately models static outlier suppression methods like affine transformations (e.g., Hadamard rotation), which essentially shift blocks to a lower crest factor $\kappa$. However, the model assumes a static absolute maximum mapping and cannot directly capture the dynamics of adaptive scale clipping methods, such as the Four-Over-Six (4/6) method (Cook et al., 2025).

## Impact Statement

This paper presents work whose goal is to offer theoretical insights about low-bits formats comparisons. There are many potential societal consequences of our work, none of which we feel must be specifically highlighted here.

## Acknowledgement

This paper is partially supported by the National Key R&D Program of China No.2022ZD0161000 and the General Research Fund of Hong Kong No.17200622 and 17209324.

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

## Outlines

- Sec. A details the proofs of Theorems 1 and 2 on INT and FP QSNR estimation.

- Sec. B presents the hardware cost estimation model.

- Sec. C provides additional details on the models used and ablation studies, and reports the numerical results corresponding to the figures in the main paper.

## A. Proofs of Theorems

### A.1. Common assumptions and notation

We consider block vectors $\mathbf{X} \in \mathbb{R}^g$ with i.i.d. entries $X_i \sim \mathcal{N}(0, \sigma^2)$. We denote the block RMS by $\sigma := \mathrm{RMS}(\mathbf{X})$ and the crest factor by

$$\kappa := \frac{\max(|\mathbf{X}|)}{\sigma}. \tag{15}$$

For MX format, which uses blockwise UE8M0 scale factors, we set

$$s' = 2^{\lceil \log_2 s \rceil} = \rho s, \qquad \rho \in [1, 2), \tag{16}$$

and choose $s' \geq s$ to avoid upper clipping. When the scale factors use BFloat16 or E4M3, we set $\rho = 1$. The ideal scale $s$ matches the largest codebook magnitude to the block maximum:

$$s = \frac{\max(|\mathbf{X}|)}{Q_{\mathrm{ref}}}, \tag{17}$$

where $Q_{\mathrm{ref}}$ depends on the target format:

- INT($b$) (symmetric): $Q_{\mathrm{ref}} = Q := 2^{b-1} - 1$ (largest integer code).

- FP($E, M, B$) (with subnormals): $Q_{\mathrm{ref}} = Q_{\max}$ (largest finite normal magnitude; e.g., $Q_{\max} = 448$ for E4M3).

This convention matches the main text: we reuse $(\sigma, \kappa, \rho, s, s')$, and $s' \geq s$ prevents overflow for both INT and FP quantization. Unless stated otherwise, expectations are over both the data and the quantization randomness, and $\|\mathbf{X}\|^2 \approx k\sigma^2$.

### A.2. Theorem 1 (INT quantization)

**INT quantization.** We consider a symmetric, uniform quantizer with bit-width $b$ and integer range $[-Q, Q]$, where

$$Q = 2^{b-1} - 1 \quad (\text{e.g., } Q \in \{127, 31, 7\} \text{ for } b \in \{8, 6, 4\}). \tag{18}$$

The quantize–dequantize operation is

$$\mathbf{X}_q = \mathrm{clamp}\big(\mathrm{round}(\tfrac{\mathbf{X}}{s'}), -Q, Q\big) \cdot s', \tag{19}$$

so the effective step in the quantization is $\Delta := s'$.

**Error model.** Let the elementwise error be $e := X - X_q$. For a non-saturating symmetric quantizer with round-to-nearest, $e \in [-\frac{\Delta}{2}, \frac{\Delta}{2}]$. Under the standard high-resolution model (Bennett, 1948), the error is approximately uniform and independent of $\mathbf{X}$:

$$\mathbb{E}[e] = 0, \qquad \mathbb{E}[e^2] = \frac{\Delta^2}{12}. \tag{20}$$

**QSNR.** Define the QSNR as

$$\mathrm{QSNR} = -10 \log_{10}\left(\frac{\|\mathbf{X} - \mathbf{X}_q\|^2}{\|\mathbf{X}\|^2}\right). \tag{21}$$

We have $\mathbb{E}[\|\mathbf{X}\|^2] \approx k\sigma^2$ and $\mathbb{E}[\|\mathbf{X} - \mathbf{X}_q\|^2] \approx k\,\mathbb{E}[e^2] = k\Delta^2/12$, hence

$$\text{QSNR} \approx -10\log_{10}\left(\frac{\Delta^2}{12\,\sigma^2}\right). \tag{22}$$

**Expressing $\Delta$ via crest factor and scale overhead.** Using Eq. (15–17),

$$\Delta = s' = \frac{\rho\,\kappa\,\sigma}{Q}. \tag{23}$$

Substituting into the QSNR expression gives

$$\frac{\Delta^2}{12\,\sigma^2} = \frac{(\rho\,\kappa)^2}{12\,Q^2}, \tag{24}$$

and therefore

$$\boxed{\text{QSNR}_{\text{MXINT}} \approx -10\log_{10}\left(\frac{\kappa^2}{12\,Q^2}\right) \approx 4.78 + 6.02\,b - 20\log_{10}(\rho) - 20\log_{10}(\kappa)} \tag{25}$$

where we use $Q \approx 2^{b-1}$ in Eq. (18). This form makes explicit: (i) $\approx 6.02$ dB per additional bit, (ii) up to 6.02 dB loss from the power-of-two overhead ($\rho \in [1, 2)$), and (iii) a penalty that scales with the crest factor $\kappa$ (which typically increases with larger block size).

**Extension to high-precision scale factors.** The analysis above assumes UE8M0 scaling, which rounds the scale and introduces the overhead $\rho \in [1, 2)$. With the E4M3 scale format used in NVINT4, the per-block scale closely matches the ideal value, so $\rho \approx 1$, and the element at the block maximum maps with (near-)zero error. For block size $g$ (elements per block), the INT QSNR with an E4M3 scale is

$$\boxed{\text{QSNR}_{\text{NVINT}} \approx -10\log_{10}\left(\frac{\kappa^2}{12\,Q^2} \cdot \frac{g-1}{g}\right) = 4.78 + 6.02\,b - 20\log_{10}(\kappa) + 10\log_{10}\left(\frac{g}{g-1}\right)} \tag{26}$$

where $10\log_{10}\left(\frac{g}{g-1}\right)$ accounts for one (near) error-free element per block.

### A.3. Theorem 2 (FP quantization)

**FP quantization.** Consider a target floating-point format $\text{FP}(E, M, B)$ with sign, $E$ exponent bits (bias $B$), and $M$ mantissa bits, with subnormals enabled. The representable numbers split into normal and subnormal domains:

$$\mathbb{C}_{\text{FP}} = \begin{cases} (-1)^s \times (1.m)_2 \times 2^{e-\text{bias}} & \text{if } e \neq 0 \text{ (Normal)}, \\ (-1)^s \times (0.m)_2 \times 2^{1-\text{bias}} & \text{if } e = 0, \, m \neq 0 \text{ (Subnormal)}, \end{cases} \tag{27}$$

where $s$, $e$, and $m$ are the sign, exponent, and mantissa of a floating-point number. Let $Q_{\max}$ denote the largest finite normal magnitude (e.g., $Q_{\max} = 448$ for E4M3), and let $N_{\min} := 2^{1-B}$ be the smallest normal. We also define the subnormal spacing in the codebook as $S_{\min} := 2^{1-B-M}$.

We use a block scale $s'$ (Eq.(16)) and perform quantize–dequantize as

$$\mathbf{X}_q = s' \cdot \text{Nearest}\left(\frac{\mathbf{X}}{s'}, \mathbb{C}_{\text{FP}}\right), \tag{28}$$

where $\mathbb{C}_{\text{FP}}$ is the FP codebook. We choose the ideal scale $s = \max(|\mathbf{X}|)/Q_{\max}$ and set $s' = \rho s$ with $\rho \in [1, 2)$ for UE8M0 (power-of-two) scaling; $\rho \approx 1$ when the scale uses E4M3.

**Error decomposition.** Let $e := \mathbf{X} - \mathbf{X}_q$. We study the relative MSE

$$R := \frac{\mathbb{E}[e^2]}{\mathbb{E}[\mathbf{X}^2]} = \frac{\mathbb{E}[e^2]}{\sigma^2}, \qquad \text{QSNR} := -10\log_{10} R. \tag{29}$$

Under a high-resolution model (Bennett, 1948), the within-cell error is unbiased and uniform on $[-\frac{\Delta}{2}, \frac{\Delta}{2}]$, and the logarithmic phase

$$r := 2^{\{\log_2(|X|/s')\}} \in [1, 2) \tag{30}$$

(the fractional part $\{\cdot\}$ of $\log_2(|X|/s')$) is approximately uniform on $[1, 2)$.

Define the signal-domain normal threshold $T_N$ and the subnormal step $\Delta_{\text{sub}}$ as

$$T_N := s' N_{\min}, \qquad \Delta_{\text{sub}} := s' S_{\min} = s' 2^{1-B-M}. \tag{31}$$

We split the amplitude axis into normal and subnormal regions:

- **Normal region** ($|X| \geq T_N$). Let $e(X) := \lfloor \log_2(\frac{|X|}{s'}) \rfloor$ be the exponent bin of $\frac{X}{s'}$. The local effective quantization step is

$$\Delta(X) = s' 2^{e(X)-M}. \tag{32}$$

Writing $2^{e(X)} = \frac{|X|}{s'r}$ with $r \in [1, 2)$ gives

$$\Delta(X) = \frac{|X|}{r} 2^{-M}. \tag{33}$$

Uniform-error modeling yields $\mathbb{E}[e^2 \mid X, |X| \geq T_N] = \frac{\Delta(X)^2}{12} = \frac{|X|^2 2^{-2M}}{12\, r^2}$. Averaging over $r \sim \text{Uniform}[1, 2]$ gives $\mathbb{E}[1/r^2] = \int_1^2 r^{-2}\, dr = 1/2$, hence

$$\mathbb{E}[e^2 \cdot \mathbf{1}\{|X| \geq T_N\}] \approx \alpha_M\, \mathbb{E}[X^2 \cdot \mathbf{1}\{|X| \geq T_N\}], \quad \alpha_M := \frac{1}{24 \cdot 2^{2M}}. \tag{34}$$

- **Subnormal but nonzero region** ($|X| < T_N$). Here the absolute spacing is constant, $\Delta_{\text{sub}}$, so

$$\mathbb{E}[e^2 \mid |X| < T_N] \approx \frac{\Delta_{\text{sub}}^2}{12} = \frac{s'^2\, 2^{2(1-B-M)}}{12}. \tag{35}$$

Let $p_{\text{sub}} := \mathbb{P}(|X| < T_N)$. Then

$$\mathbb{E}[e^2 \cdot \mathbf{1}\{|X| < T_N\}] \approx \frac{s'^2\, 2^{2(1-B-M)}}{12}\, p_{\text{sub}}. \tag{36}$$

Summing the two contributions and normalizing by $\sigma^2$ yields

$$\frac{\mathbb{E}[e^2]}{\sigma^2} \approx \alpha_M\, w_{\text{norm}} + \beta\, (\rho\,\kappa)^2\, p_{\text{sub}}, \tag{37}$$

where we define the dimensionless weight

$$w_{\text{norm}} := \frac{\mathbb{E}[X^2 \cdot \mathbf{1}\{|X| \geq T_N\}]}{\sigma^2}, \tag{38}$$

and use $\frac{s'^2}{\sigma^2} = \frac{(\rho\kappa)^2}{Q_{\max}^2}$ with

$$\beta := \frac{2^{2(1-B-M)}}{12\, Q_{\max}^2}. \tag{39}$$

Therefore,

$$\boxed{\text{QSNR}_{\text{MXFP}} \approx -10 \log_{10}\big(\alpha_M\, w_{\text{norm}} + \beta\, (\rho\,\kappa)^2\, p_{\text{sub}}\big)} \tag{40}$$

In the ample dynamic-range regime ($w_{\text{norm}} \approx 1$ and $p_{\text{sub}} \approx 0$), the law simplifies to

$$\text{QSNR} \approx -10 \log_{10}(\alpha_M) = 13.80\,\text{dB} + 6.02\,M\,\text{dB}, \tag{41}$$

independent of block granularity and the distribution of $\mathbf{X}$.

**Extension to high-precision scale factors.** The analysis above assumes a UE8M0-quantized scale, which forces $s'$ to be a power of two and introduces the overhead $\rho \in [1, 2)$. When the per-block scale uses E4M3 (as in NVFP4), the scale closely tracks the ideal value, so $\rho \approx 1$, and the element at the block maximum maps with negligible error (its scaled value hits $Q_{\max}$). It is therefore natural to exclude the block-maximum contribution from the normal-region error budget. Let $g$ be the block size and define the energy fraction of the block maximum as

$$\eta := \frac{\max(|\mathbf{X}|)^2}{g\,\sigma^2} = \frac{\kappa^2}{g}. \tag{42}$$

Setting $\rho = 1$ and replacing $w_{\mathrm{norm}}$ by $w_{\mathrm{norm}} - \eta$ in Eq. (40) yields the refined QSNR approximation for FP quantization with an E4M3 scale:

$$\boxed{\mathrm{QSNR}_{\mathrm{NVFP}} \approx -10\log_{10}\!\big(\alpha_M\,(w_{\mathrm{norm}} - \tfrac{\kappa^2}{g}) + \beta\,\kappa^2\,p_{\mathrm{sub}}\big)} \tag{43}$$

This adjustment isolates the block maximum and tightens the prediction when the scale is represented with sufficient precision.

*Table 6.* Gate-complexity model for the MAC Unit with $k$ lanes. Here $x$ and $y$ denote exponent and mantissa widths; for INT, $x{=}0$. The aligner width $n$ is given by (44). "Main Cells" list dominant standard cells used in aggregation.

| Sub-block | INT Mul | FP Mul | INT Add | FP Add | Main Cells |
|---|---|---|---|---|---|
| Multiplier | $k(x+y+1)^2$ | $k(y+1)^2$ | – | – | AND, FA, HA |
| Adder (mantissa/int) | – | – | $2k(x+y+1)$ | $kn$ | FA, HA |
| Exponent adder | – | $kx$ | – | – | FA, HA |
| Exponent subtractor | – | – | – | $kx$ | XOR, FA, HA |
| Comparator | – | – | – | $kx$ | XOR, AND, OR |
| Aligner (barrel) | – | – | – | $k\,n\log_2 n$ | MUX |
| Normalizer (shared) | – | – | – | $n\log_2 n$ | MUX, OR |

# B. Hardware Cost Modeling

**Scope and assumptions.** We develop a compact gate-level model to estimate the chip area and energy of a GEMM engine under low-precision formats. Specifically, a low-bit GEMM engine uses four components: a quantizer, a multiply-and-accumulate (MAC) unit, a dequantizer, and an FP32 accumulator. The proposed model accounts only for the MAC unit, a shared FP32 accumulator and a dequantizer; the quantizer is excluded from all cost accounting. In MX/NV formats, the VPU implements quantization by shift/divide-and-round, and the accumulation pipeline can fuse dequantization as two 8-bit integer additions for UE8M0 scale or two floating-point multiplications for E4M3 scale. We omit the quantizer block in VPU to isolate the cost driven by multiplication and accumulation. Unless otherwise stated, we take cell factors from a TSMC FinFET standard-cell library. We model only combinational logic; we ignore sequential elements, placement and routing, and interconnect to enable technology-aware, relative comparisons.

| Throughput Ratio | INT8 : INT4 = 1 : 2 |
|---|---|
| No reuse | 1 * int8_MAC_unit + 2 * int4_MAC_unit |
| INT reuse scheme 1 | 1 * int8_MAC_unit + 1 * int4_MAC_unit |
| INT reuse scheme 2 | 2 * int8_(u)int4_MAC_unit |
| Throughput Ratio | FP8 : FP4 = 1 : 2 |
| No reuse | 1 * e4m3_MAC_unit + 2 * e2m1_MAC_unit |
| FP reuse scheme | 1 * e4m3_MAC_unit + 1 * e2m1_MAC_unit |

*Table 7.* Comparison of MAC unit configurations with the same lanes for different reuse schemes. Notes: (1) No reuse: Highest energy efficiency for INT8 and INT4, but greatest area wastage; (2) INT reuse scheme 1: Use int8 lane as an int4 path directly (set the 8-b input to XXXX_0000), a little more energy cost for INT4 but lower area cost; (3) INT reuse scheme 2: Use two int8×(u)int4 lanes to reconfigure int8 lane or int4 lane, a little more energy cost for both INT4 and INT8, but lowest area cost; (4) No reuse: Highest energy efficiency for FP8 and FP4, but greatest area wastage; (5) FP reuse scheme: Use fp8 lane as an fp4 path directly (set the 8-b input to S_00XX_X00), a little more energy cost for FP4 but lower area cost. We adopt INT reuse scheme 2 and FP reuse scheme to evaluate the area cost shown in Table 5.

**Design choice: FP32 accumulation and MMU integration.** A high-throughput Matrix-Multiply Unit (MMU), as in TPU-like designs (Norrie et al., 2021), integrates the multiply-and-accumulate datapath and downstream accumulation to improve performance and energy efficiency. To prevent error growth and preserve scalability, we accumulate in FP32. Under the same nominal bit width, FP multipliers are typically more area- and energy-efficient than INT multipliers, whereas FP adders are more expensive than INT adders due to exponent comparison/subtraction, mantissa alignment, and normalization (Zhang et al., 2024b). With a uniform-alignment design (Ul Haq et al., 2025), the normalizer count reduces to one shared instance across the $k$ MAC lanes, and we divide its cost by $k$.

**Mantissa aligner width.** The mantissa aligner couples accuracy and cost: its bit width $n$ affects numerical fidelity and hardware complexity. We set

$$n = \min(2^{x+1} + 2y, \texttt{psum\_bit\_width}), \tag{44}$$

where $x$ and $y$ denote exponent and mantissa widths, respectively (for INT formats, $x=0$). In all evaluations we use $k=32$ for MX formats and $k=16$ for NV formats, and $\texttt{psum\_bit\_width}=24$.

**MAC unit structure and sub-blocks.** We model the MAC unit as a $k$-lane array. Each lane comprises one multiplier. The adders from all lanes are fused together to form a multi-input adder tree structure, incorporating FP-specific alignment and normalization logic. Table 6 reports the dominant logic count (up to constant factors) for the main sub-blocks, where "Main Cells" indicate the standard-cell types used for area/energy aggregation. For FP multiplication, we multiply only mantissas and include an exponent adder. For FP addition, we model exponent comparator/subtractor, a barrel aligner, a wide mantissa adder, and one shared normalizer. For INT, we set $x=0$ in the expressions.

**Area and energy aggregation for MAC.** Let $\mathcal{S}=\{$Multiplier, Adder(mantissa/int), Exponent adder, Exponent subtractor, Comparator, Aligner(barrel), Normalizer(shared)$\}$ be the set of sub-block types, and $\mathcal{G} = \{$FA, HA, XOR, AND, OR, MUX$\}$ be the set of cell types with technology-dependent area and energy factors $A_g$ and $E_g$ obtained from the standard-cell library. Let $\tau_g$ be the toggle rate of cell $g$, which represents the average switching activity of the cell. In this work, we simplify the toggle rate factor by assuming that all gate cells have the same toggle rate, $\tau_g = \tau$, to reduce computational complexity and focus on the primary design trade-offs. Denote by $c_{s,g}(x,y,k,n)$ the count of cell $g \in \mathcal{G}$ in sub-block $s$ induced by the chosen format and by $n$ from Eq.(44). The MAC area and energy are

$$\text{Area}_{\text{MAC}} = \sum_{s\in\mathcal{S}}\sum_{g\in\mathcal{G}} c_{s,g}(x,y,k,n)\, A_g, \qquad \text{Energy}_{\text{MAC}} = \sum_{s\in\mathcal{S}}\sum_{g\in\mathcal{G}} c_{s,g}(x,y,k,n)\, E_g\tau_g. \tag{45}$$

**FP32 accumulator model.** We model the FP32 accumulator by its combinational logic counts $c_g^{\text{ACC32}}$, yielding

$$\text{Area}_{\text{ACC32}} = \sum_{g\in\mathcal{G}} c_g^{\text{ACC32}}\, A_g, \qquad \text{Energy}_{\text{ACC32}} = \sum_{g\in\mathcal{G}} c_g^{\text{ACC32}}\, E_g\tau_g. \tag{46}$$

**Dequantizer model.** We model the shared dequantizer based on the logic required for the specific format (e.g., fused integer additions or floating-point multiplications as described in §B). We aggregate its combinational logic counts $c_g^{\text{DEQ}}$, yielding

$$\text{Area}_{\text{DEQ}} = \sum_{g\in\mathcal{G}} c_g^{\text{DEQ}}\, A_g, \qquad \text{Energy}_{\text{DEQ}} = \sum_{g\in\mathcal{G}} c_g^{\text{DEQ}}\, E_g\tau_g. \tag{47}$$

**Total cost and per-lane reporting.** The total MMU cost is

$$\begin{aligned}
\text{Area}_{\text{MMU}} &= \text{Area}_{\text{MAC}} + \text{Area}_{\text{DEQ}} + \text{Area}_{\text{ACC32}}, \\
\text{Energy}_{\text{MMU}} &= \text{Energy}_{\text{MAC}} + \text{Energy}_{\text{DEQ}} + \text{Energy}_{\text{ACC32}},
\end{aligned} \tag{48}$$

and, when we report per-lane figures, we divide the cost of shared blocks (the dequantizer and the FP32 accumulator) by $k$.

**Summary.** The hardware model includes the MAC unit, the dequantizer, and the FP32 accumulator; the quantizer is excluded from the overhead calculation. Given a low-precision format with exponent/mantissa widths $(x,y)$ (with $x=0$ for INT), a MAC array size $k$, an aligner cap $\texttt{psum\_bit\_width}$ (setting $n$ via Eq (44), and technology cell factors $\{A_g, E_g\}_{g\in\mathcal{G}}$ (plus the dequantizer and FP32-accumulator gate counts), the model predicts the area and energy of the MAC and accumulation stages. It captures the relative cost trends across MX/NV-INT/FP formats at the same nominal bit width, the sensitivity to the aligner width $n$ (critical for FP addition), and the effect of sharing both the normalizer, the dequantizer, and the FP32 accumulator across $k$ lanes.

# C. More Details for Reproduction

## C.1. Used Models

*Table 8.* Huggingface IDs of evaluation models in direct-cast inference.

| Model Name | Huggingface ID |
|---|---|
| Qwen3-0.6B | Qwen/Qwen3-0.6B-Base |
| Qwen3-1.7B | Qwen/Qwen3-1.7B-Base |
| Qwen3-4B | Qwen/Qwen3-4B-Base |
| Qwen3-8B | Qwen/Qwen3-8B-Base |
| Qwen3-14B | Qwen/Qwen3-14B-Base |
| Qwen3-32B | Qwen/Qwen3-32B |
| Qwen3-30B-A3B | Qwen/Qwen3-30B-A3B-Instruct-2507 |
| Qwen3-235B-A22B | Qwen/Qwen3-235B-22B-Instruct-2507 |
| Llama-3.2-1B | meta-llama/Llama-3.2-1B |
| Llama-3.2-3B | meta-llama/Llama-3.2-3B |
| Llama-3.1-8B | meta-llama/Meta-Llama-3.1-8B |
| Llama-3.1-70B | meta-llama/Meta-Llama-3.1-70B |

**Models for inference evaluation.** We list the Huggingface IDs of evaluated open-sourced model for better reproduction in Table 8. Note that we firstly choose the base model without supervise fine-tuning if it is open-sourced. For a model of a certain size, our selection principle is that if the base model is open source, we will first choose the base model; otherwise, we will select the model that has undergone SFT.

*Table 9.* Llama-3 style Model architecture and training hyper-parameters.

| Model Size | 145M | 1B | 3B |
|---|---|---|---|
| Layers | 12 | 16 | 28 |
| Hidden Size | 1024 | 2048 | 3072 |
| FFN Hidden Size | 3072 | 8192 | 8192 |
| Attention Heads | 16 | 32 | 24 |
| KV Heads | 4 | 8 | 8 |
| Batch Size (# Sequence) | 256 | 512 | 512 |
| Max LR | 1.0e-3 | 6e-4 | 6e-4 |
| Min LR | | $0.1 \times$ Max LR | |
| Optimizer | | AdamW ($\beta_1 = 0.9, \beta_2 = 0.95$) | |
| Weight Decay | | 0.1 | |
| Clip Grad Norm | | 1.0 | |
| LR Schedule | | Cosine | |
| Warmup Steps | | 500 | |
| Sequence Length | | 2048 | |

**Models for training evaluation.** We select the Llama-3 (Dubey et al., 2024) style model for our experiments due to its wide adoption. The Llama-3 style model employs Group Query Attention (GQA)(Ainslie et al., 2023) for the self-attention module and SwiGLU(Shazeer, 2020) for the feed-forward module. Table 9 presents the detailed architectural settings and training hyper-parameters of the models used.

## C.2. Necessity of Symmetric Integer Representation

Table 10 offer the ablation studies on representation range of INT8 quantization. We find that the bias in representation range would consistently degenerate INT8 training loss. For BFloat16 scale factor, we can find that asymmetric representation

*Table 10.* Ablation studies about the clipping range on INT8 quantization across quantization granularities, as well as BFloat16 and UE8M0 scale factors. We report the 8-bit training loss (lower is better) on a 145M model with 20B training tokens. The baseline of BF16 training without quantization

| | BF16 scale | | UE8M0 scale | |
| --- | --- | --- | --- | --- |
| | [-128, 127] | **[-127, 127]** | [-128, 127] | **[-127, 127]** |
| per-channel | **3.2544** | 3.2560 | **3.3602** | **3.4307** |
| 256 | 3.1340 | **3.1307** | 3.1628 | **3.1574** |
| 128 | 3.1309 | **3.1289** | 3.1353 | **3.1326** |
| 64 | 3.1312 | **3.1269** | 3.1312 | **3.1288** |
| 32 | 3.1354 | **3.1251** | 3.1299 | **3.1269** |

range even making block 32 quantization worse than block 256 quantization. This is because only the minimal values in each quantization block have possibility to be quantized into 128 in INT8 quantization, and smaller block size indicates more individual quantization blocks. Additionally, asymmetric quantization also causes degeneration for UE8M0 scale factors, but the degeneration strength is slighter than BFloat16 scales. This is because UE8M0 scale factor consistently greater than or equal to Bfloat16 scale, leading less high-precision number to map to $Q_{min}$. These experiments demonstrate the necessity of symmetric representation space for integer quantization.

---

**Algorithm 1** Analyzing Numerical Stability of Different Floating-Point Precisions

---

1: **Input:** Dimension $N = 4096$, precision list $P = \{\text{bfloat16}, \text{float16}, \text{float32}\}$
2: **Output:** Ratio of elements equal to 128 for each precision
3: **for** each *precision* in $P$ **do**
4:     $D \leftarrow \text{GenerateRandomMatrix}(N, N, \text{precision})$                                    // Generate $N \times N$ on GPU
5:     $S \leftarrow D/127$
6:     $D_{\text{norm}} \leftarrow \text{Round}(D \oslash S)$
7:     $count \leftarrow \text{CountElementsEqualTo}(D_{\text{norm}}, 128)$
8:     $total \leftarrow N \times N$
9:     $ratio \leftarrow count/total$
10:    **print** "Precision:", *precision*, ", Ratio:", $ratio$
11: **end for**

---

*Table 11.* Results of Algorithm 1.

| BFloat16 | Float16 | Float32 |
| --- | --- | --- |
| 16.82% | 0.02% | 0 |

**Numerical stability analysis.** We also analyze the numerical stability of different float-point for quantization mapping through Algorithm 1. Table 11 shows the results of Algorithm 1, demonstrating that in BFloat16 precision, a significant portion of values (16.82%) are mapped to -128. This phenomenon occurs even though the scaling factor s is theoretically designed to map the value to 127. In conclusion, this analysis highlights a critical pitfall of using low-precision floating-point formats for quantization calculations. The inherent lack of precision in bfloat16 and, to a lesser extent, float16 can lead to overflow during the scaling step, incorrectly mapping values to outside the intended integer range. This powerfully demonstrates that a forced symmetric clipping step is essential for guaranteeing the correctness and stability of quantization, particularly when the computation is performed using low-precision data types.

## C.3. Detailed Results

This section offer detailed numbers of experiments, as follows:

- Table 12 and Table 13 present the KL divergence results, corresponding to Table 3.

- Table 14 and Table 15 present the perplexity results, for better understanding the relationship between KL divergence and perplexity. They are consistent in most case.

*Table 12.* Qwen3 models KL divergence (lower is better) results across different low-bit formats in direct-cast inference. All reported KL metrics are the average over all tokens, multiplied by $10^6$.

| Format | 0.6B | 1.7B | 4B | 8B | 14B | 32B | 30B-A3B | 235B-A22B |
|---|---|---|---|---|---|---|---|---|
| | | | | Qwen-3 | | | | |
| MXINT8 | **191** | **209** | **112** | **168** | **96** | **118** | **160** | **276** |
| MXFP8 | 579 | 406 | 346 | 362 | 300 | 457 | 380 | 483 |
| MXINT6 | 1944 | 2464 | 928 | 1104 | 804 | 1012 | 768 | 1333 |
| MXFP6 | **1030** | **874** | **539** | **592** | **467** | **627** | **606** | **1099** |
| MXINT4 | 39936 | 30208 | 17408 | 15552 | 34304 | 27392 | 13248 | 16331 |
| MXFP4 | **17602** | **14614** | **8568** | **8228** | **8119** | **10302** | **6194** | **16238** |
| NVINT4 | 10560 | 8320 | 4864 | 5120 | 5568 | 7968 | 3120 | 9702 |
| NVFP4 | **8104** | **4995** | **3844** | **3430** | **2835** | **3778** | **2443** | **9238** |
| | | | Qwen-3 **(w/ random Hadamard rotation)** | | | | | |
| Format | 0.6B | 1.7B | 4B | 8B | 14B | 32B | 30B-A3B | 235B-A22B |
| MXINT8 | **137** | **150** | **80** | **130** | **70** | **88** | **135** | **229** |
| MXFP8 | 921 | 1321 | 468 | 577 | 393 | 497 | 391 | 707 |
| MXINT6 | 1137 | **1274** | 547 | 690 | 481 | 615 | 444 | 809 |
| MXFP6 | **1007** | 1446 | **497** | **618** | **454** | **558** | **422** | **740** |
| MXINT4 | 26488 | 26578 | 10498 | 12241 | 8459 | 9510 | 6080 | 9660 |
| MXFP4 | **17995** | **20443** | **7260** | **8562** | **6410** | **6536** | **5087** | **7058** |
| NVINT4 | **7771** | **7236** | **3431** | **4026** | **3070** | **3647** | **2222** | **3931** |
| NVFP4 | 12031 | 10582 | 5065 | 5912 | 4214 | 4662 | 3200 | 5786 |

*Table 13.* Llama-3 models KL divergence (lower is better) results across different low-bit formats in direct-cast inference. All reported KL metrics are the average over all tokens, multiplied by $10^6$.

| Format | 3.2-1B | 3.2-3B | 3.1-8b | 3.1-70B |
|---|---|---|---|---|
| | | Llama | | |
| MXINT8 | **111** | **77** | **82** | **191** |
| MXFP8 | 464 | 325 | 359 | 514 |
| MXINT6 | 1133 | 743 | 776 | 1744 |
| MXFP6 | **651** | **457** | **491** | **1436** |
| MXINT4 | 26153 | 14089 | 12380 | 22538 |
| MXFP4 | **14446** | **8251** | **7586** | **21372** |
| NVINT4 | 7508 | 4312 | 4224 | 10970 |
| NVFP4 | **5691** | **3684** | **3718** | **10544** |
| | Llama**(w/ random Hadamard rotation)** | | | |
| Format | 3.2-1B | 3.2-3B | 3.1-8b | 3.1-70B |
| MXINT8 | **89** | **63** | **65** | **145** |
| MXFP8 | 573 | 388 | 409 | 1393 |
| MXINT6 | 773 | 531 | 558 | 1518 |
| MXFP6 | **643** | **447** | **457** | **1476** |
| MXINT4 | 20126 | 11116 | 10272 | 137612 |
| MXFP4 | **11967** | **8269** | **7189** | **129471** |
| NVINT4 | **5854** | **3912** | **3609** | **19975** |
| NVFP4 | 8129 | 5240 | 4752 | 77363 |

*Table 14.* Qwen3 models perplexity (lower is better) results of WikiText2 across different low-bit formats in direct-cast inference.

| | Qwen-3 | | | | | | | |
|---|---|---|---|---|---|---|---|---|
| Format | 0.6B | 1.7B | 4B | 8B | 14B | 32B | 30B-A3B | 235B-A22B |
| BF16 | 11.5868 | 8.7084 | 7.3368 | 6.5135 | 5.9498 | 7.0168 | 6.8178 | 4.0929 |
| MXINT8 | **11.6377** | **8.7424** | **7.3511** | **6.5174** | **5.955** | **7.0185** | **6.8167** | **4.0959** |
| MXFP8 | 11.7494 | 8.7822 | 7.3813 | 6.5444 | 5.9711 | 7.0357 | 6.8335 | 4.1101 |
| MXINT6 | 12.2297 | 9.2622 | 7.496 | 6.6499 | 6.0483 | 7.05 | 6.8745 | 4.1743 |
| MXFP6 | **11.9108** | **8.8961** | **7.4135** | **6.5825** | **5.9953** | **7.0285** | **6.8467** | **4.1662** |
| MXINT4 | 48.6713 | 21.8749 | 11.9487 | 10.0423 | 16.7227 | 15.1619 | 9.3837 | 5.918 |
| MXFP4 | **20.4522** | **24.0766** | **9.1553** | **8.0135** | **7.2471** | **8.2047** | **7.8203** | **5.9007** |
| NVINT4 | 15.9729 | 10.9128 | 8.3304 | 7.415 | 6.81 | 8.0161 | 7.2024 | 4.8916 |
| NVFP4 | **14.6818** | **9.9966** | **8.0144** | **7.0285** | **6.3129** | **7.3604** | **7.1874** | **4.8309** |
| | Qwen-3**(w/ random Hadamard rotation)** | | | | | | | |
| Format | 0.6B | 1.7B | 4B | 8B | 14B | 32B | 30B-A3B | 235B-A22B |
| MXINT8 | **11.6179** | **8.7240** | **7.3407** | **6.5170** | **5.9521** | **7.0187** | **6.8231** | **4.0973** |
| MXFP8 | 11.8629 | 8.9972 | 7.4068 | 6.5898 | 5.9839 | 7.0448 | 6.8918 | 4.1287 |
| MXINT6 | **11.9422** | **9.0122** | **7.4071** | 6.6119 | 5.9905 | **7.0627** | **6.8666** | **4.1263** |
| MXFP6 | 11.9096 | 9.0089 | 7.4108 | **6.5911** | **5.9981** | 7.0787 | 6.8711 | 4.1252 |
| MXINT4 | **28.6510** | **21.3032** | **9.8238** | **9.2029** | **7.3564** | **8.2083** | **7.8292** | **4.9891** |
| MXFP4 | 20.3684 | 15.9527 | 8.8148 | 8.1113 | 6.9521 | 7.7401 | 7.9673 | 4.7035 |
| NVINT4 | **14.6052** | **10.7822** | **7.9824** | **7.1705** | **6.3702** | **7.3625** | **7.1557** | **4.3913** |
| NVFP4 | 16.5762 | 11.7541 | 8.2716 | 7.5084 | 6.5427 | 7.4522 | 7.3214 | 4.5918 |

*Table 15.* Llama-3 models perplexity (lower is better) results of WikiText2 across different low-bit formats in direct-cast inference.

| | Llama | | | |
|---|---|---|---|---|
| Format | 3.2-1B | 3.2-3B | 3.1-8b | 3.1-70B |
| BF16 | **9.0625** | **7.2857** | **5.8402** | **2.637** |
| MXINT8 | **9.0815** | **7.2944** | **5.8487** | **2.6674** |
| MXFP8 | 9.1695 | 7.3381 | 5.895 | 2.6674 |
| MXINT6 | 9.3557 | 7.4184 | 5.9643 | 2.7298 |
| MXFP6 | **9.2209** | **7.3605** | **5.916** | **2.7298** |
| MXINT4 | 21.9893 | 11.2715 | 8.7408 | 5.1894 |
| MXFP4 | **14.0516** | **9.2355** | **6.4845** | **4.9492** |
| NVINT4 | 11.3987 | 8.225 | 6.5957 | 3.5502 |
| NVFP4 | **10.7473** | **8.0343** | **6.4917** | **3.492** |
| | Llama**(w/ random Hadamard rotation)** | | | |
| Format | 3.2-1B | 3.2-3B | 3.1-8b | 3.1-70B |
| MXINT8 | **9.0715** | **7.2912** | **5.845** | **2.6428** |
| MXFP8 | 9.1932 | 7.3465 | 5.9001 | 2.7232 |
| MXINT6 | 9.2622 | 7.3828 | 5.9276 | 2.7333 |
| MXFP6 | **9.2204** | **7.3703** | **5.9075** | **2.735** |
| MXINT4 | 17.9797 | 10.3057 | 8.0745 | 1146.7256 |
| MXFP4 | **13.3987** | **9.262** | **7.2318** | **1118.4431** |
| NVINT4 | **10.8399** | **8.1119** | **6.4701** | **4.9786** |
| NVFP4 | 11.7635 | 8.4693 | 6.7028 | 79.7586 |

