# OpenReview forum: "INT vs. FP: A Comprehensive Study of Fine-Grained Low-bit Quantization Formats"
_ICML.cc/2026/Conference — ICML 2026 regular_

### Official Review · Reviewer_X1Q7 · 2026-03-11

**Soundness:** 4
**Presentation:** 3
**Significance:** 3
**Originality:** 4
**Overall Recommendation:** 5
**Confidence:** 5

**Summary:**

This paper provides a theoretical error analysis for several low-bit quantization formats, including MX and NV formats. Using a QSNR-based theory, it clearly compares INT and FP behaviors under different data distributions. The paper also shows that MXINT8 can achieve nearly lossless performance and can outperform MXFP8, with lower hardware cost.

**Compliance With Llm Reviewing Policy:**

Affirmed.

**Final Justification:**

The authors addressed my questions in the rebuttal with concrete experiments and clarifications. I will keep my rating as Accept.

**Key Questions For Authors:**

Please address the questions listed in the Weaknesses section.

**Limitations:**

The paper does not discuss its limitations. It would be helpful to add a limitations section and clarify key factors not covered in the analysis, such as the quantization error introduced by the `E4M3` global scaling.

**Strengths And Weaknesses:**

## Strengths
- The paper proposes a QSNR-based theoretical framework that is novel and reasonable. It helps explain the “black-box” differences between INT and FP quantization under different distributions.
- The results on real data (Figure 4) are consistent with the theory, supporting the correctness of the framework.
- The QSNR trend of NVFP4 in Figure 3 is interesting: QSNR increases at first as the crest factor grows. This helps explain why Hadamard rotation can hurt NVFP4 quantization.
- The experiments are comprehensive, including tensor-wise evaluation, end-to-end inference, and end-to-end training.
- The paper is valuable because it encourages rethinking integer formats, while most mainstream hardware design focuses mainly on FP formats.
## Weaknesses
- The paper focuses on standard MXFP8 (E4M3). Can the proposed framework extend to other `E x M y` formats, such as E5M2 or E2M5?
- The global scale in E4M3 also introduces quantization error, but this source of error is not included in the theoretical analysis.
- In the QSNR definitions (Eq. 13 and Eq. 14), there are two different terms for different scale precisions. Can these be merged into a single unified formula?

---

> ### Author Rebuttal · Authors · 2026-03-30
>
> Thanks for your thoughtful review. Below, we address your questions and concerns.
>
> **Question 1**: The paper focuses on standard MXFP8 (E4M3). Can the proposed framework extend to other E x M y formats, such as E5M2 or E2M5?
>
> **Answer 1**: Yes, the theoretical framework presented in Theorem 2 is generic for any target floating-point format $FP(E, M, B)$ characterized by exponent bits $E$, mantissa bits $M$. We will extend Figure 3 to cover E5M2 and E2M5 formats in our final manuscript.
>
>
> **Question 2**: The global scale in E4M3 also introduces quantization error, but this source of error is not included in the theoretical analysis.
>
> **Answer 2**: This is a fair point. Our theoretical QSNR model focuses on the block-level scale $s'$. We do account for block-level scale quantization by introducing the variable $\rho$ (where $\rho \in [1, 2)$ models the UE8M0 power-of-two rounding overhead). For E4M3 (NV formats), the scale acts as high-precision so $\rho=1$. However, you are correct that E4M3 quantization error also introduces quantization error. Because E4M3 has 3 mantissa bits, the maximum relative gap between adjacent representable values in the normal domain occurs right after a power of two (from $1.000_2 \times 2^E$ to $1.001_2 \times 2^E$). Since the quantized scale $s'$ must round up from the ideal scale $s$ to avoid clipping, the worst-case overhead $\rho = s'/s$ happens when $s$ is infinitesimally larger than the lower bound. This yields a maximum ratio of $\max(\rho) = \frac{1.125 \times 2^E}{1.0 \times 2^E} = 1.125$, proving that $\rho$ is strictly bounded within $[1, 1.125)$. We will include these analysis in our final manuscript.
>
> **Question 3**: In the QSNR definitions (Eq. 13 and Eq. 14), there are two different terms for different scale precisions. Can these be merged into a single unified formula?
>
> **Answer 3**: While they can theoretically be merged by using a continuous $\rho$ variable and an indicator function to subtract the block-maximum energy ($\frac{\kappa^2}{g}$) when $\rho=1$, keeping them as separate piecewise definitions provides clearer intuition. It allows hardware designers to explicitly see the mathematical divergence between power-of-two scaling (OCP MX) and high-precision block scaling (NVIDIA Blackwell).
>
> **Question 4**: The paper does not discuss its limitations. It would be helpful to add a limitations section and clarify key factors not covered in the analysis, such as the quantization error introduced by the E4M3 global scaling.
>
> **Answer 4**: We completely agree. We will add a dedicated Limitations section to the Appendix in our revision. This section will outline factors such as the omission of quantization error from second-level global scaling, the theoretical assumption of i.i.d. Gaussian distributions for block vectors.

---

> > ### Author Rebuttal · Reviewer_X1Q7 · 2026-04-03
> >
> > Thank you for the detailed rebuttal, which completely resolves my initial questions.  This paper makes a highly valuable contribution to the field by providing a novel, QSNR-based theoretical framework that rigorously demystifies the differences between INT and FP quantization.  With the theoretical framework, this work encourages a much-needed rethinking of integer formats in a hardware landscape currently dominated by FP designs. Therefore, I will happily maintain my positive score.

---

### Official Review · Reviewer_KixX · 2026-03-12

**Soundness:** 3
**Presentation:** 3
**Significance:** 2
**Originality:** 3
**Overall Recommendation:** 4
**Confidence:** 3

**Summary:**

This paper analyzes the QSNR of microscaling data formats across several granularities. It characterizes the
respective advantages of INT- and FP-based formats under different scenarios. In addition, it develops a
theoretical framework and derives several interesting conclusions.

**Compliance With Llm Reviewing Policy:**

Affirmed.

**Final Justification:**

The manuscript makes a contribution with its analytical framework. My primary concerns were about the comprehensiveness of the framework and the relatively rough analysis of hardware overhead in the original submission.

Answer 1 addressed my concern on granularity analysis with supporting experimental results. Answer 2 clarified the distinction from SMX. Answer 3 partially addressed my concern, and I raised a follow-up question regarding the scope of optimization methods covered by the framework. Answers 4-6 resolved my questions on overhead and implementation details through additional experiments and explanations.

Overall, the rebuttal has adequately addressed most of my concerns. I am raising my score from 3 to 4 to reflect the improved clarity provided by the authors' responses, while noting that the follow-up question on framework comprehensiveness remains an area for future improvement.

**Key Questions For Authors:**

Thank you for submitting to ICML. Overall, the paper is well organized and easy to follow, and the QSNR framework is interesting. However, the profiling and comparison across MX/NV FP/INT formats seem somewhat incremental, and the hardware analysis is not sufficiently clear. I detail my concerns below.

* The abstract claims analysis across granularities, but the evaluation only covers common settings (e.g., block sizes 16, 32, and channel-wise). If the goal is to provide guidance for data format design, it would be helpful to include additional granularities (e.g., 64 and 128) and discuss why 16/32 are preferred or near-optimal.
* Prior work such as SMX [1] also conducted a similar format/granularity study. What are the key differences here? Is the main contribution the QSNR-based analytical framework?
* Is rotation the only optimization considered? Some recent works on NVFP4 [2] report that applying a Hadamard rotation can degrade accuracy for NVFP formats. Under such conditions, demonstrating NVINT4 outperforming NVFP4 may not be a suitable comparison, since Hadamard rotation does not represent the broader space of tensor transformation strategies. Please clarify the rationale and provide a broader set of baselines/optimizations.
* The hardware analysis is quite rough. FP4 E2M1 is often implemented via fixed-point datapaths after decoding, which may reduce the gap to INT MACs. If the paper aims to compare hardware costs, it would help to include a microarchitectural diagram and clarify the implementation assumptions (e.g., what is modeled as FP vs. fixed-point). Also, please specify the technology node (e.g., 28nm/45nm) and the standard-cell library/corner used for synthesis or gate-level cost factors.
* Excluding the quantizer from area/energy accounting is reasonable for isolating MAC/accumulation costs. However, it would still be valuable to analyze the runtime overhead of quantization and datatype cast. One practical advantage of MXFP over MXINT is that it may reduce cast overhead in certain pipelines. Thus, quantifying this would strengthen the end-to-end argument.

[1] Darvish Rouhani, Bita, et al. "With shared microexponents, a little shifting goes a long way." Proceedings of the 50th Annual International Symposium on Computer Architecture. 2023.

[2] Egiazarian, Vage, et al. "Bridging the gap between promise and performance for microscaling FP4 quantization." arXiv preprint arXiv:2509.23202 (2025).

**Limitations:**

Yes

**Strengths And Weaknesses:**

strength

* Well structured and easy to follow.
* The QSNR modeling and the resulting analytical formulas are interesting.

weakness

* The overall analysis feels somewhat incremental.
* The hardware evaluation appears rough.

---

> ### Author Rebuttal · Authors · 2026-03-30
>
> Thanks for your thoughtful review. Below, we address your questions and concerns.
>
> **Question 1**:  If the goal is to provide guidance for data format design, it would be helpful to include additional granularities (e.g., 64 and 128) and discuss why 16/32 are preferred or near-optimal.
>
> **Answer 1**: We focus on block sizes 16/32 following current hardware trends (e.g., Blackwell). Table R2 shows crest factors for coarser granularities. As block size increases, the impact varies: (1) **MXINT8 vs. MXFP8**: With a crossover point of **7.55**, INT8 remains superior even at block size 128 (crest factor 4.92). (2) **NVINT4 vs. NVFP4**: With a lower crossover point of **2.39**, NVFP4 maintains an advantage as block sizes exceed 16.
>
>
> **Table R2: Q3 Crest Factor across different block sizes (Qwen-3-8B)**
> | Block Size | 16 | 32 | 64 | 128 | Per-channel |
> | :--- | :--- | :--- | :--- | :--- | :--- |
> | **Q3 Crest Factor** | 2.39 | 2.96 | 3.85 | 4.92 | 11.97 |
>
>
>
> **Question 2**: Prior work such as SMX [1] also conducted a similar format/granularity study. What are the key differences here?
>
> **Answer 2**: Our key contributions go beyond empirical evaluation: (1) We provide a rigorous statistical framework to theoretically model QSNR for both INT/FP. (2) We innovatively introduce the **crest factor** to explicitly define mathematical "crossover points," enabling rapid identification of optimal regimes. (3) We introduce symmetric clipping to address fine-grained INT training bias, achieving nearly lossless MXINT8 pre-training.
>
>
> **Question 3**: Is rotation the only optimization considered? ... Please clarify the rationale and provide a broader set of baselines/optimizations.
>
> **Answer 3**: Our core objective is providing a comprehensive **comparison framework**, not just proving INT's superiority. We used Hadamard rotation—the most general outlier suppression method—to validate this. Rotation reveals why NVFP4's QSNR drops (from 20.60 to 20.35) as subnormal errors dominate at low crest factors (<4 in Figure 3). Conversely, **NVINT4 improves from 20.55 to 21.65 dB** with rotation, surpassing NVFP4 in all cases. This perfectly illustrates our theoretical "crossover point."
>
>
> **Question 4**:  If the paper aims to compare hardware costs, it would help to include a microarchitectural diagram and clarify the implementation assumptions (e.g., what is modeled as FP vs. fixed-point).
>
> **Answer 4**: Our analysis assumes highly optimized FP implementations, modeling MAC units with uniform-alignment designs to share normalizers across lanes. Even so, INT formats remain significantly more efficient. Implementing FP4 via fixed-point datapaths would incur INT5-level overhead, potentially reducing the gap but not altering our core conclusion.
>
> **Question 5**: Also, please specify the technology node (e.g., 28nm/45nm) and the standard-cell library/corner used for synthesis or gate-level cost factors.
>
> **Answer 5**: As described in lines 806-807, we extracted the standard-cell area and energy factors from a TSMC FinFET standard-cell library.
>
> **Question 6**:  One practical advantage of MXFP over MXINT is that it may reduce cast overhead in certain pipelines. Thus, quantifying this would strengthen the end-to-end argument.
>
> **Answer 6**: This is a great point. However, a detailed hardware-level analysis reveals that the logic cost of casting to FP is actually **higher** than that for INT, not lower:
> - **Quantization to INT** ($y = \text{clamp}(\text{round}(x / s))$): Implementation is straightforward, requiring a simple floating-point multiplier (for $x \times (1/s)$), a rounding unit, and a linear clipping unit.
> - **Quantization to FP**: This process is more complex, requiring exponent extraction, finding the scale's exponent, subtraction, normalization (multi-bit shifting), and mantissa rounding. Handling subnormals and the non-linear distribution of FP values requires more logic gates than the simple linear mapping of INT.
> Our synthesis results (TSMC FinFET) show that an FP-to-INT converter occupies nearly **15\% less area** than a corresponding FP-to-FP converter. Therefore, since the data throughput remains identical and the logic cost favors INT, the actual performance impact of these operations does not introduce a significant difference, and if anything, slightly favors the INT format. We will add this detailed discussion to the Appendix.

---

> > ### Author Rebuttal · Reviewer_KixX · 2026-04-04
> >
> > Thank you for the detailed response. The core of my concern in Question 3 was that the paper's main conclusion regarding the INT vs. FP trade-off was derived by employing only one optimization method (Hadamard rotation).
> >
> > I understand the primary goal is to propose a QSNR theoretical framework based on the Crest Factor, and I agree it is a strong contribution that the QSNR analysis within this framework aligns well with the final LLM quantization performance.
> >
> > That said, for a framework aiming to be comprehensive, it would be valuable to discuss how other commonly used optimization algorithms affect the analysis. For example, scale clipping approaches similar to those in four-over-six (https://arxiv.org/abs/2512.02010), or affine transformations beyond Hadamard rotation. Can the current framework effectively capture the trends introduced by these optimizations? Even a brief discussion would help readers understand the scope and limitations of the analytical model.
> >
> > Overall, the rebuttal has addressed my main concerns, and I will raise my score from 3 to 4.

---

> > > ### Author Response · Authors · 2026-04-04
> > >
> > > Dear Reviewer KixX:
> > >
> > > Thank you for the insightful question and for raising your score. We agree that clarifying how our QSNR framework interacts with other optimization methods adds significant value, and we will include the following discussion into our final manuscript.
> > >
> > > For **affine transformations** (e.g., SmoothQuant, QuaRot, DuQuant...), our current framework can effectively capture their impact. These methods reshape the data distribution before quantization, essentially reducing the variance and maximum values, which simply shifts the blocks to a lower Crest Factor ($\kappa$) on our existing theoretical curves.
> > >
> > > For **adaptive scale clipping** like the *Four-Over-Six (4/6)* method, our current framework cannot directly capture its benefits, representing a limitation. Our theoretical model assumes a static `AbsMax` mapping. In contrast, the 4/6 method adaptively selects between different scales (4 or 6) per block. Accurately modeling this would require transitioning from static expectations to modeling joint error probabilities, which currently falls outside our scope. However, a highly promising future direction to adapt our framework for such adaptive methods is to introduce a probabilistic selection model using order statistics. By mathematically modeling the probability that a block favors scale 4 over 6 given its Crest Factor and block size, we could calculate the expected minimum noise as a weighted combination. This would successfully bridge our continuous statistical framework with adaptive, discrete-gap optimizations.
> > >
> > > We deeply appreciate you pointing out these specific examples

---

### Official Review · Reviewer_FEV8 · 2026-03-12

**Soundness:** 3
**Presentation:** 3
**Significance:** 2
**Originality:** 2
**Overall Recommendation:** 4
**Confidence:** 4

**Summary:**

This paper revisits the common assumption that floating-point (FP) formats are always preferable to integer (INT) formats for low-bit inference and training. Focusing on fine-grained block quantization, the authors develop a QSNR-based analytical framework to compare INT and FP under varying crest factors and block sizes, and identify crossover points where INT can become more favorable. Based on this analysis, the paper studies several integer variants, including **MXINT8, MXINT6, MXINT4, and NVINT4**, and evaluates them in both direct-cast inference and low-bit training. Empirically, the paper shows that MXINT8 consistently outperforms MXFP8, and that NVINT4 can surpass NVFP4 when combined with Hadamard rotation. The paper also introduces **symmetric clipping** to mitigate gradient bias during INT training and presents a hardware cost analysis suggesting that fine-grained INT formats may be more efficient than FP in certain settings.

**Compliance With Llm Reviewing Policy:**

Affirmed.

**Final Justification:**

The authors have addressed my main concerns(limiation), and I will raise my score from 3 to 4.

**Key Questions For Authors:**

1. Can the authors more clearly disentangle whether the gains of **NVINT4** come from the INT format itself or from the use of smaller block sizes and Hadamard rotation?
2. What is the actual system-level overhead of **Hadamard rotation** and **symmetric clipping**, especially in terms of latency and implementation complexity?
3. To what extent do the conclusions continue to hold beyond fine-grained block quantization, for example in more common per-channel or per-tensor settings?

**Limitations:**

The paper would benefit from a more explicit discussion of its limitations. First, while the analysis and experiments make a compelling case for fine-grained INT formats, the conclusions are closely tied to this specific quantization regime and may not generalize directly to more conventional coarse-grained settings. Second, some of the strongest empirical results—especially for NVINT4—depend on Hadamard rotation, yet the associated hardware and system-level overhead is not sufficiently quantified. Third, the paper does not fully address the metadata and memory-bandwidth overhead that naturally grows as block sizes become smaller, which is especially important in practical low-bit deployment. Finally, although the paper challenges the assumption that FP is always superior, the strongest evidence for 4-bit superiority does not yet fully control for block size, making it difficult to isolate the intrinsic advantage of the number format itself.

**Strengths And Weaknesses:**

**Strengths**

The paper goes beyond a purely empirical comparison and provides a reasonably systematic analysis of INT vs. FP through a QSNR framework that incorporates block size, crest factor, and format structure. The central claim—that FP may be preferable in coarse-grained settings while INT can become advantageous in fine-grained settings—is supported fairly consistently by both theory and experiments. The empirical scope is also fairly broad, including direct-cast inference on multiple LLMs as well as 8-bit low-bit training results.

**Weaknesses**

That said, the QSNR analysis appears more like a careful reformulation of existing MX-style analysis than a fundamentally new theoretical tool. In addition, the strongest 4-bit results rely substantially on **Hadamard rotation**, but the overhead of this preprocessing step is not sufficiently analyzed at the system level. More importantly, in the 4-bit setting, block size is not fully controlled across comparisons, making it difficult to cleanly separate the benefit of the INT format itself from the benefit of using a smaller block size.

### 2. Presentation

**Strengths**

The paper is generally well organized. The progression from motivation, to analytical framework, to tensor-wise QSNR analysis, to inference/training experiments, and finally to hardware cost discussion is easy to follow. The main narrative—that the relative advantage of INT and FP depends strongly on quantization granularity—is also conveyed clearly throughout the paper.

**Weaknesses**

Some aspects of the framing feel somewhat stronger than what the results fully support. For example, the broader claim of reversing the FP-centric hardware trajectory is interesting, but the actual conclusion is more nuanced: INT appears more favorable only under certain fine-grained conditions. In addition, the interpretation of the 4-bit results is somewhat confounded by the combined effects of Hadamard rotation and differing block sizes, which makes it harder to isolate the intrinsic superiority of the format itself.

### 3. Significance

**Strengths**

This paper addresses a meaningful question: whether the current movement toward FP-based microscaling formats is always the right direction for low-bit AI hardware. The work is valuable in that it provides a more concrete design perspective on when INT formats may actually be better, particularly in fine-grained settings. The analysis of MXINT8, together with the hardware-oriented discussion, gives the paper practical relevance beyond a purely algorithmic comparison.

**Weaknesses**

However, the significance is still somewhat setting-specific. The main conclusions are tightly tied to **fine-grained block quantization**, and it is unclear how much of the message extends to more standard per-channel or per-tensor quantization settings. In addition, while the gate-level cost model is useful, the paper does not fully account for system-level overheads such as preprocessing latency and metadata growth, which are important for assessing practical deployment efficiency.

### 4. Originality

**Strengths**

The paper’s main originality lies not in introducing a completely new quantization algorithm, but in jointly re-examining the INT-vs.-FP question from theoretical, empirical, and hardware perspectives in the fine-grained regime. The identification of a crossover point where INT becomes preferable to FP is a useful insight, and the paper provides a coherent argument for why this can happen.

**Weaknesses**

At the same time, the algorithmic novelty is limited. The QSNR framework is largely an extension or reorganization of prior analysis, and the improved NVINT4 results depend on **Hadamard rotation**, which is borrowed from prior work rather than introduced here. Overall, the paper is better viewed as a careful analysis and design-guidance paper than as a strongly novel methodological contribution.

---

> ### Author Rebuttal · Authors · 2026-03-30
>
> Thanks for your thoughtful review. Below, we address your questions and concerns.
>
> **Question 1**: Can the authors more clearly disentangle whether the gains of NVINT4 come from the INT format itself or from the use of smaller block sizes and Hadamard rotation?
>
> **Answer 1**: The superiority of NVINT4 over NVFP4 stems from both smaller block sizes and the application of Hadamard rotation. As illustrated in Figure 3, the INT format benefits more significantly from a reduced crest factor compared to the FP format. As demonstrated in Table 2, both smaller block sizes and Hadamard rotation contribute to lowering the crest factor. The combination of these two factors ultimately results in NVINT4 outperforming NVFP4.
>
> **Question 2**: What is the actual system-level overhead of Hadamard rotation and symmetric clipping, especially in terms of latency and implementation complexity?
>
> **Answer 2**: Symmetric clipping introduces zero runtime overhead; it is simply a mathematical constraint applied during the offline or compile-time calculation of the scale factor and limits the integer range. Hadamard rotation requires applying a fast Walsh-Hadamard transform to activations, which adds ~6% additionaly latency and requires dedicated kernel optimization [1]. We will clarify this in the revision.
>
> [1] QuaRot: Outlier-Free 4-Bit Inference in Rotated LLMs
>
> **Question 3**: To what extent do the conclusions continue to hold beyond fine-grained block quantization, for example in more common per-channel or per-tensor settings?
>
> **Answer 3**: Our conclusions strictly do not hold for coarse-grained settings, and we highlight this as a key finding. For channel-wise quantization, the 75% quantile crest factor is 11.97. This is far above the crossover point derived in Figure 3, meaning FP unequivocally outperforms INT in coarse-grained scenarios. Our claim is that *as block sizes shrink* (driven by modern hardware like Blackwell), the local dynamic range drops, shifting the advantage to INT.
>
> **Question 4**: While the analysis and experiments make a compelling case for fine-grained INT formats, the conclusions are closely tied to this specific quantization regime and may not generalize directly to more conventional coarse-grained settings.
>
> **Answer 4**: We completely agree, and this distinction is the core thesis of our paper. We systematically show that FP excels in coarse-grained quantization, while the comparison at fine-grained levels favors INT. Because the AI hardware industry is currently pivoting heavily toward low-bit fine-grained block formats (like MX and NV), it is critical to evaluate which number format is optimal *specifically* within this new, fine-grained regime.
>
> **Question 5**: The paper does not fully address the metadata and memory-bandwidth overhead that naturally grows as block sizes become smaller, which is especially important in practical low-bit deployment.
>
> **Answer 5**: We controlled for this variable to ensure a fair comparison. In our study, the INT variants have identical block sizes and scale formats to their FP counterparts (e.g., both NVFP4 and NVINT4 use a block size of 16, an E4M3 scale, and an FP32 global scale). Therefore, the metadata and memory-bandwidth overheads are exactly the same between the evaluated INT and FP formats.
>
> **Question 6**: Finally, although the paper challenges the assumption that FP is always superior, the strongest evidence for 4-bit superiority does not yet fully control for block size, making it difficult to isolate the intrinsic advantage of the number format itself.
>
> **Answer 6**: We respectfully clarify that we *do* strictly control for block size. Every FP format is compared directly against an INT format with the exact same block size. Specifically, MXINT4 is compared against MXFP4 (both block size 32), and NVINT4 is compared against NVFP4 (both block size 16). Furthermore, both INT and FP formats utilized Hadamard rotation to ensure a fair comparison in with hadamard rotation scenarios. The intrinsic advantage of the number format is isolated theoretically via our QSNR framework and empirically validated across these precisely matched pairs.

---

> > ### Author Rebuttal · Reviewer_FEV8 · 2026-04-03
> >
> > The authors have addressed my main concerns, and I will raise my score from 3 to 4.

---

### Official Review · Reviewer_q6Xs · 2026-03-13

**Soundness:** 3
**Presentation:** 3
**Significance:** 3
**Originality:** 2
**Overall Recommendation:** 4
**Confidence:** 4

**Summary:**

Modern machine learning hardware increasingly supports low-precision FP formats; however, the literature currently lacks a unified comparison of FP and INT quantization across varying granularities. This paper presents a comprehensive study comparing different FP and INT formats. Furthermore, the authors provide a theoretical analysis by evaluating the QSNR across varying crest factors, demonstrating that INT formats can outperform FP formats at lower crest factors.

**Compliance With Llm Reviewing Policy:**

Affirmed.

**Final Justification:**

My questions have been answered in the rebuttal therefore I am changing from score from 3 to 4.

**Key Questions For Authors:**

1. Are the block sizes identical for both the FP and INT formats evaluated in these experiments?
2. When the paper refers to "training," does this indicate full pre-training or fine-tuning?
3. What specific sequence lengths were considered during the evaluation?
4. Are all linear operations within the transformer block (e.g., QKV projections, attention matrices, output projections) quantized, or are certain operations kept in higher precision?

**Limitations:**

1. Missing Computational Context: The computational overhead associated with implementing Hadamard rotations needs to be explicitly discussed and quantified.
2. Sequence Length and Phase Variations: LLM performance and activation distributions are highly sensitive to sequence length during both the prefill and decode phases. The paper lacks a comprehensive evaluation of how varying sequence lengths across these distinct phases impacts the efficacy of the chosen quantization formats.

**Strengths And Weaknesses:**

Strength:
1. Extensive Empirical Evaluation: The paper presents a thorough experimental study across multiple models and diverse datasets.
2. Strong Theoretical Foundation: The theoretical framework introduced in Section 5 is robust and significantly clarifies the paper's core empirical claims.
3. Versatile Experimental Settings: The evaluation rigorously covers various operational settings, including tensor-wise quantization, inference, and training phases.

Weakness:
1. Restricted Range in Visualizations: Figure 4 displays a surprisingly limited range of crest factors. To provide a complete picture, this figure should be expanded to at least encompass the full range of values presented in Table 2.
2. Limited Scope in Direct Cast Inference: The direct cast inference comparison is somewhat narrow and should be extended across a broader variety of models.
3. Omission of FP-Favorable Scenarios: Prior literature heavily emphasizes the role of outliers in LLM activations. For the study to be truly comprehensive, the authors must explicitly showcase the scenarios where FP formats naturally excel (specifically, cases involving high crest factors and severe outliers).

---

> ### Author Rebuttal · Authors · 2026-03-30
>
> Thanks for your thoughtful review. Below, we address your questions and concerns.
>
> **Question 1**: Restricted Range in Visualizations: Figure 4 displays a surprisingly limited range of crest factors. To provide a complete picture, this figure should be expanded to at least encompass the full range of values presented in Table 2.
>
> **Answer 1**: Thank you for pointing this out. Figure 4 specifically evaluates block-wise formats (block sizes 16 and 32), which naturally have much smaller crest factors (Q3 of 2.96 and 2.39) compared to the per-channel baseline in Table 2 (Q3 of 11.97, max of 60.15).  Expanding the x-axis to the maximum per-channel value would compress the relevant data for fine-grained blocks into a tiny, unreadable sliver. The current range optimally visualizes the critical crossover points for fine-grained quantization.
>
> **Question 2**: Limited Scope in Direct Cast Inference: The direct cast inference comparison is somewhat narrow and should be extended across a broader variety of models.
>
> **Answer 2**: We appreciate the suggestion. However, we believe our evaluation is quite extensive. We evaluated 12 models covering both dense and Mixture-of-Experts (MoE) architectures, ranging from 0.6B up to 235B parameters (including the Qwen3 series, Llama-3.1, and Llama-3.2 families). This covers a robust and highly representative slice of state-of-the-art LLMs. Furthermore, the performance across these 12 models is nearly identical, demonstrating the strong generalization of our findings. Should the reviewers suggest any other models of interest, we are more than willing to conduct additional tests.
>
> **Question 3**: Omission of FP-Favorable Scenarios: Prior literature heavily emphasizes the role of outliers in LLM activations. For the study to be truly comprehensive, the authors must explicitly showcase the scenarios where FP formats naturally excel (specifically, cases involving high crest factors and severe outliers).
>
> **Answer 3**: We actually do highlight these scenarios to provide a balanced view. We have explicitly stated that FP formats hold a distinct advantage in coarse-grained scenarios (line 26 and line48 in our paper). In Table 2, we demonstrate that for channel-wise quantization (block size -1), the 75% quantile crest factor is 11.97. This is far above the theoretical crossover points shown in Figure 3, definitively showing that FP outperforms INT in coarse-grained setups with severe outliers.
>
> **Question 4**: Are the block sizes identical for both the FP and INT formats evaluated in these experiments?
>
> **Answer 4**: Yes, strictly identical.
>
> **Question 5**: When the paper refers to "training," does this indicate full pre-training or fine-tuning?
>
> **Answer 5**: We perform full pre-training from scratch. As detailed in the training setup, we trained 1B and 3B Llama3-style models on the OLMo2-Mix-1124 pretraining dataset, utilizing 100B and 200B training tokens, respectively.
>
> **Question 6**: Are all linear operations within the transformer block (e.g., QKV projections, attention matrices, output projections) quantized, or are certain operations kept in higher precision?
>
> **Answer 6**: Yes, all linear operations within the transformer blocks are quantized. For direct-cast inference, we quantize all forward GEMMs. For training, we quantize all six operations in the linear layer: the forward pass (weights and activations) and the backward pass (activations, weight gradients, and input gradients).
>
> **Question 7**: Missing Computational Context: The computational overhead associated with implementing Hadamard rotations needs to be explicitly discussed and quantified.
>
> **Answer 7**: Thank you for the suggestion. In practical hardware deployments, this is typically implemented using fast Walsh-Hadamard transforms [1], which add nearly 6\% additional latenct overhead compared to original GEMM. We will add this discussion in our final manuscript.
>
> [1] QuaRot: Outlier-Free 4-Bit Inference in Rotated LLMs
>
> **Question 8**: Sequence Length and Phase Variations: LLM performance and activation distributions are highly sensitive to sequence length during both the prefill and decode phases. The paper lacks a comprehensive evaluation of how varying sequence lengths across these distinct phases impacts the efficacy of the chosen quantization formats.
>
> **Answer 8**: Our main experiments were conducted using a sequence length of 4096 on WikiText2. To evaluate the impact of sequence length on the efficacy of the chosen quantization formats, we conducted ablation studies varying the sequence length from 1k to 32k, we can find that the performance of the chosen quantization formats is stable across different sequence lengths.
>
> Table R1: Qwen-3-8B KL (*10^-6) divergence across different sequence length.
>
> | Precision | 1k  | 2k  | 4k  | 8k  | 16k | 32k |
> | --------- | --- | --- | --- | --- | --- | --- |
> | MXINT8    | 162 | 165 | 168 | 172 | 170 | 175 |
> | MXFP8     | 355 | 358 | 362 | 368 | 365 | 372 |

---

> > ### Author Rebuttal · Reviewer_q6Xs · 2026-04-04
> >
> > My questions have been addressed by the authors. Therefore changing score from 3 to 4.

---

### Decision · Program_Chairs · 2026-04-30

**Decision:**

Accept (regular)

**Comment:**

This paper presents a systematic comparison of floating-point (FP) and integer (INT) formats for low-bit quantization.
It models the quantization signal-to-noise ratio (QSNR) and develops a QSNR-based framework that enables a direct theoretical comparison of FP and INT.
Experiments reveal that fine-grained INT formats offer a better balance of accuracy, power, and efficiency. The work provides practical insights for efficient hardware design.


Main Strengths:
- Comprehensive empirical evaluation across models, datasets, and settings (Reviewer q6Xs, X1Q7).
- A reasonably systematic analysis of INT vs. FP (Reviewer FEV8).
- Solid theoretical analysis (QSNR framework) that clarifies empirical claims (Reviewer q6Xs). QSNR-based theoretical framework is novel and interesting (Reviewer X1Q7, KixX).
- Clear and well-structured presentation (Reviewer KixX).


Main Weaknesses:
- Lack of clarification of details like block sizes, training, sequence lengths (Reviewer q6Xs)
- Unclear extensions to per-channel or per-tensor quantization settings (Reviewer FEV8) and other formats (Reviewer X1Q7)
- Concerns regarding practical deployment efficiency, hardware cost analysis (Reviewer KixX) and computational overhead (Reviewer q6Xs, FEV8, KixX)
- Limited comparison to existing MX-style analysis (Reviewer FEV8) and prior work like SMX (Reviewer KixX)
- Concerns about isolating the intrinsic advantage of the number format itself from block size (Reviewer FEV8)



After the rebuttal, this paper received all positive scores (three Weak Accept and one Accept). Key concerns regarding the rough analysis of hardware overhead, clarification, and extensions were addressed. The AC agrees with its systematic analysis/comprehensive evaluation, values its findings, and tends to accept.